# Cause-effect discovery in Hydrometeorological Systems: Evaluation of Causal Discovery methods.

Vivek Kumar Yadav<sup>1,2</sup>, Murray Peel<sup>2</sup>, Keirnan Fowler<sup>2</sup>, Dongryeol Ryu<sup>2</sup>, and Bramha Dutt Vishwakarma<sup>1</sup>

Correspondence: Vivek Kumar Yadav (viveky@iisc.ac.in)

**Abstract.** Identifying the driver(s) of a process or phenomenon is central to understanding and predicting its future state. In complex hydrometeorological systems, a process can have multiple drivers dynamically coupled to the system across timescales. Thus, a robust method to identify drivers is imperative. In hydrological sciences, methods like multivariate regression and, more recently, Big Data machine-learning approaches rely on finding a co-relation between variables, rather than identifying cause-effect relations. This study evaluates cause-effect discovery (Causal Discovery or CD) algorithms in hydrometeorological systems. Although earlier studies have made important contributions to exploring CD methods, they have primarily focused on bivariate methods in simple synthetic environments. Specifically, we evaluate the following four theoretically distinct multivariate CD algorithms, (i) TCDF (ii) VARLiNGAM, (iii) PCMCI+, and (iv) DYNOTEARS. We evaluate these algorithms within a large, complex simulated environment of the Global Land Data Assimilation System (GLDAS) where the drivers, reference truth, are known perfectly. We evaluate the drivers identified by CD methods against this reference truth and also contrast its results with the widely used method of co-relation identification, Pearson's Correlation Coefficient (PCC). The results show that CD methods identify fewer false drivers compared to PCC, across a range of Köppen-Geiger climate types. For example, PCC failed to distinguish true drivers from instantaneous and lagged cross-correlations, typically present in hydrometeorological systems. Whereas, CD methods eliminate a higher number of false instantaneous and lagged drivers. Thus, though PCC identifies the highest number of true drivers, it suffers from high false drivers. Overall, CD methods perform similar to or better than PCC, while PCMCI+ and DYNOTEARS performed the best. Further, we test whether time-series prediction models perform better when predictors are limited to those identified as causal by CD methods. Evaluation of surface soil moisture predictions during drought shows that CD-based models outperform PCC-based models and are more parsimonious. Thus, we demonstrate the effectiveness of using causal discovery to eliminate spurious relations and obtain a robust set of drivers for prediction and process understanding across different climate conditions. This study overviews, demonstrates and tests efficacy of CD methods in studying cause-effect relations in hydrometeorological systems. By exposing their capabilities and differences in a simulated environment, we hope to encourage their use in the real world and move beyond co-relation.

<sup>&</sup>lt;sup>1</sup>Interdisciplinary Centre for Water Research, Indian Institute of Science, Bengaluru, 560012, India

<sup>&</sup>lt;sup>2</sup>Faculty of Engineering and Information Technology, The University of Melbourne, Melbourne 3010, Victoria, Australia

#### 1 Introduction

The Earth's hydrological system is a complex system of energy, water, and nutrient circulation. It interacts, at various spatial and temporal scales, with weather, climate and human interventions. Changes in the system via change in the state of variables and their interaction patterns cause diverse events such as floods, droughts, heatwaves and changes in streamflow regimes. To understand, adapt and mitigate such events, or for sustainable use of water resources, a comprehensive understanding of the processes leading to such phenomena is required. Fundamental to process understanding is a robust method to identify the true drivers of a process (Christian et al., 2024; Van Oldenborgh et al., 2022; Barriopedro et al., 2023; Mishra et al., 2022). Historically, driver identification has been based on correlation, simple regression, and probability-based models such as multivariate regression, auto-regressive modelling, and combinations thereof (Tasker, 1980; Holder, 1985). These methods rely on maximising correlation or lagged-correlation, rather than identifying direct causation.

Over time, process understanding has been translated into models of the hydrological system that have grown in complexity with increasing availability of data and computational resources (Peel and McMahon, 2020). These physically based models encode process understanding and drivers into numerical schemes to simulate hydrological variables (Beven et al., 1984; Beven, 1989; Zhang and Montgomery, 1994; Warszawski et al., 2014; Dutra et al., 2017), with several of these models now simulating the water cycle of the Earth (Schellekens et al., 2017; Gosling et al., 2023). However, these models are limited by approximations and parametrisations within their governing equations to represent sub-grid and sub-timestep processes, which has led research attention towards purely data-driven methods of Machine Learning and Artificial Intelligence (ML) to improve model performance (Zhang et al., 2018; Kratzert et al., 2019; Nearing et al., 2021; Feng et al., 2023).

While ML methods can outperform physical models (Kratzert et al., 2019; Xu and Liang, 2021; Nearing et al., 2021), they replace physically-based process understanding with complex and opaque model architectures. Despite large volumes of data being required to train these models (Tripathy and Mishra, 2024), methods like Random Forest, Long Short-Term Memory and Decisions Trees are growing in popularity in hydrology. Although ML methods can provide outstanding results, the relational

approach at their core, by definition, falls short of identifying causal predictors. This results in a two-fold problem. First, it prohibits identifying drivers of a process, which is critical to decipher the impact of climate change on the water cycle across spatio-temporal scales. Second, is the old problem of "getting the right answers for the wrong reasons" (Kirchner, 2006). This is evident when modellers interpret ML results; they rely on predictor importance methods to explain plausible model structures, which do not capture cause-effect relations. This task is challenging due to the opaque nature and complexity of ML

structures, which do not capture cause-effect relations. This task is challenging due to the opaque nature and complexity of MI methods (Nearing et al., 2021; Samek et al., 2019; Höge et al., 2022).

An alternative approach to identify drivers is to make use of the considerable progress that has been made in the science of cause and effect. Causal discovery (CD) is concerned with finding cause-effect (Causal) relations among variables from purely observational data, while causal inference offers methods to quantify the effect of intervening in a system and quantifying the influence of certain variables, using CD or actual interventional data (Pearl, 2009; Peters et al., 2017). So far, only a few studies have used CD for studying hydrometeorological systems. Following is a short summary of key terms, the Granger Causality (GC), introduced by Granger (1969), uses statistical measures to find causality between a pair of variables. Specifically, if

activities.

including the past of a variable X, reduces the residuals of a prediction of Y, then X Granger causes Y. The Transfer Entropy (TE) (Schreiber, 2000), is an information theoretic extension of GC that finds the difference in information contained in a variable Y, with or without a given variable X, where the measure of information is the Shannon Entropy (Shannon, 1948). Convergent Cross Mapping (CCM), introduced by Sugihara et al. (2012), is a method based on time-delay embedding and reconstruction of deterministic dynamical systems to determine causality between a pair of variables. Finally, Pearl's Causality (Pearl, 1998, 2009), uses Graphs (Bayesian Networks) to represent the causal relations of a multivariate system, like PC-alg (Spirtes and Glymour, 1991). PC-alg uses conditional independence tests to find causal parents (drivers) of each variable in the system. For a brief history of the development of CD methods we suggest reading Ombadi et al. (2020).

In hydro-meteorology applications of CD have primarily used Granger Causality based methods, bi-variate methods, approaches that do not account auto-correlation, or methods based on deterministic dynamical theory. Examples include Ruddell and Kumar (2009), who used TE to find causality between ecohydrological processes during different seasons. Tuttle and Salvucci (2017) used GC to understand the effect of precipitation persistence and seasonality in soil-moisture and precipitation feedback. Rinderer et al. (2018) used GC, TE and various measures of correlation and information flow to understand subsurface hydrologic connectivity. Goodwell et al. (2020) used various information theoretic measures to identify different types of plausible interactions in a multivariate system. Wang et al. (2018) used CCM to explore the effect of soil moisture on precipitation. Similarly, Bonotto et al. (2022) used CCM to find causality between groundwater and streamflow and reported weaker causal links during and after a drought period. Delforge et al. (2022) used CCM and graphical modelling based PCMCI (an extension of PC-alg) to discover hydrologic connectivity in a synthetic and real karstic site. Shi et al. (2022) used CCM to eliminate the spurious bi-directional correlation between meteorological and hydrological drought indices, isolate the causality from meteorological to hydrological drought, and estimate drought propagation times. Chauhan et al. (2023) used PCMCI

to discover the interconnections of hydrologic and thermodynamic fluxes across neighbouring basins. While Wang et al. (2025) used PCMCI to understand the causal interactions in a complex system comprising ecological, hydrological and human

Hydrological time-series are typically stochastic, multivariate, highly interconnected, contain self-causation (via auto-correlation) and contemporaneous causal relations. CD methods capable of handling such systems are required to unravel the true causality in Hydrological Sciences. While the adoption of CD methods in Hydrological Sciences is growing, it has been limited predominantly to GC, TE and CCM. Ombadi et al. (2020) provides an example where four CD methods, GC, TE, CCM, and PC-alg, were evaluated on the output of a simple bucket hydrological model and reported their results in the context of noise, time series length, and sample size. Several of these methods have limitations, particularly in the context of hydrological sciences. For example, GC, TE, and CCM are bivariate methods and cannot find the correct causation where a third (or more) variable acts as a confounder (common driver) between two variables (Ombadi et al., 2020; Delforge et al., 2022). Hydrometeorological systems are typically highly interconnected, across different timescales, with multiple variables responsible for driving a process. Similarly, many variables show strong state dependence (self-causation via autocorrelation) which cannot be handled by GC, TE, CCM or PC-alg (Runge et al., 2019b). Further, certain causal interactions happen at contemporaneous times. Since GC, TE by definition look for causal relations from past to future values, they cannot handle contemporaneous

100

105

interactions (Granger, 1969; Sugihara et al., 2012), while PC-alg also does not consider contemporaneous interactions (Runge, 2022). Finally, real-world observations of hydrological systems are typically noisy and contain uncertainties. The deterministic dynamical system assumption of CCM limits its use in such cases (Sugihara et al., 2012; Ombadi et al., 2020).

In this study, we extend the evaluation of CD methods in a complex hydrometeorological system by evaluating four theoretically distinct methods of causal discovery. The algorithms overviewed and evaluated use frameworks suitable to find causal relations in multivariate time-series data. Further, by considering auto-correlation and cross-lagged and contemporaneous relations, these are suitable to identify self causation and causal relations across multiple time lags. Finally, by not assuming a deterministic system, these are theoretically well suited to the stochastic nature of hydrometeorological systems. Developed across diverse contexts and problems, we evaluate the following CD algorithms: *i*) Score-based structure learning: DYNOTEARS (Pamfil et al., 2020), *ii*) Noise-based: VARLiNGAM (Hyvärinen et al., 2008), *iii*) Constraint-based method: PCMCI+ (Runge, 2022), and *iv*) Granger causality based: Temporal Causal Discovery Framework (Nauta et al., 2019). We evaluate their performance by their ability to identify known causal drivers within a simulated dynamical system, GLDAS 2.0 (Li et al., 2018). We seek to answer the following questions:

- a) Can CD methods identify the true drivers in a complex simulated hydrometeorological system?
- b) What is their overall performance, in terms of identifying causal relations and eliminating non-causal co-relations?
- c) What is the trade off between choosing a correlation-based approach and CD methods?
- d) Identifying causal drivers of hydrological variables and time-series predictions.

The primary aim of this paper is to overview, demonstrate and evaluate state-of-the-art methods of Causal Discovery for identifying true drivers of a process. By reviewing the causal discovery literature, we select methods better suited for hydrometeorological systems, and apply them on a large and complex simulated environment to recover the process drivers. Then, we contrast the results with PCC to expose the redundancies introduced by relying on correlation based methods. Further, to understand the significance of finding causal drivers in applications, we demonstrate its use to obtain parsimonious models for robust prediction under changing conditions. Like Ombadi et al. (2020), we hope this work encourages the hydrology community to embrace Causal Discovery methods for robust and interpretable understanding and transcend beyond the limitations of *co*-relation based approaches.

The paper is organised as follows: Section 2 lays out the approach for evaluating CD methods while describing some representations of causality and explains the CD methods evaluated here. Section 3 presents the results of overall evaluation across different climate zones and in particular of causal drivers of surface soil moisture. In Section 4 we evaluate the performance of each CD method, provide some perspectives towards applying them, and discuss the limitations of our work. Section 5 summarises main findings of this work.

# 2 Methodology and Methods

We divide this section into two parts: *Methodology*, which lays out the overall approach for the analysis and *Methods*, which explains the CD methods, their assumptions and their evaluation strategy adopted. We begin the Methodology subsection with a summary of the overall methodology adopted to evaluate the performance of CD methods. In the Methods sub-sections we describe some standard concepts and methods to represent cause-effect relations. Then we describe some metrics to evaluate different methods based on these representations. We present details of our synthetic environment and the resulting reference truth for evaluating the methods. Next, we describe the CD methods evaluated here, followed by a detailed explanation of each method and their assumptions. Finally, we describe the strategy adopted to test the efficacy of CD-based time-series prediction models.

# 2.1 Overall Methodology

The evaluation of CD methods was conducted in a simulated environment, since discovering true causal relationships from real-world observational data is inherently challenging. Several factors can complicate both the application and interpretation of CD methods: mismatches between the timescales of processes and their observations, the presence of observational or process noise, or simply the unavailability of key variables of a process. Even when such difficulties are absent, establishing causality for well-understood processes remains a non-trivial task (Delforge et al., 2022; Ombadi et al., 2020). Applying CD methods in a synthetic environment avoids such issues. Although these simulated environments are only abstractions of the real world, they provide the crucial benefit of knowing the true causal relations via their generating equations.

To evaluate the ability of CD methods to discover true causal relations from data, we applied them on output from a physics-based hydro-meteorological Land Surface Model. Based on a literature review of the model structure and its governing equations (Appendix A) we determined which variables are causally related, which formed the reference truth against which the causal methods could be compared (Fig. 2a). Then, we applied the CD methods to the simulated data and recorded the estimated causal relations. These estimated causal relations were then compared to the reference truth to evaluate the performance of each CD method.

Detecting the presence of causal links is important to understand the connections between various processes. Similarly, correctly identifying the absence of causal links is important to eliminate correlated yet causally unrelated variables. This provides a parsimonious picture and potentially leads to a simpler representation of the overall system. Thus, our evaluation involves accuracy of CD methods on both aspects, of correctly identifying the causal links and their absence.

Further, to understand the robustness of CD methods in different climatic conditions, we performed the analysis on data from nine different locations. These sites span eight distinct Köppen-Geiger climate classes across Tropical, Temperate, Arid, and Cold zones. We also compare our results with a non-causal method, we conducted the analysis using the Pearson's Correlation Coefficient (PCC) method as well. We selected PCC due to its simple interpretation and wide acceptance.

Finally, we demonstrate the application of causal knowledge acquired above to a typical problem in hydrology, that of split-sample prediction. We apply PCC and CD methods as a predictor selection step to identify the predictors of surface soil

moisture. We feed these predictor sets into machine learning models for predicting surface soil moisture time-series and evaluate their performances under drought period. The next section describes methods to represent causal relations in a multivariate system.

### 2.2 DAG and Adjacency Matrix

Figure 1. Three equivalent methods to describe cause-effect relationship. In a) variables  $X_t^1, X_t^3$  and  $X_t^1, X_t^4$  are input variables to generate  $X_t^2$  and  $X_t^3$  respectively. This is represented as a graph in b) with directed edges from nodes  $X_t^1, X_t^3$  into node  $X_t^2$  and from nodes  $X_t^1, X_t^4$  into node  $X_t^3$ . The adjacency matrix in c) represents this with binary operators in the corresponding cell of two variables, for example the directed edge in b) between  $X_t^4$  and  $X_t^3$  is shown with '1' in the fourth column (cause) and third row (effect). d) shows an exemplar adjacency matrix compared and its cells classified, with respect to c).

In a dynamical system, output variables change state through forcing variables applied to the system, and in response to coupling amongst variables, boundary conditions, thresholds and process noise. Consider the simple dynamical system in Fig. 1a where the cause-effect (causal) relation among variables is represented by functional relationships. This causal relationship can be schematised using graphs as well (Fig. 1b). Graphs represent the relations between variables (nodes) using arrows or links (edges). To represent a causal relation with a graph, it requires two necessary conditions, directed edges and acyclicity.

Since causal relations are direct cause-effect relations, a causal graph requires all the edges to be directed. Further, due to temporal ordering of cause-effect relations, a causal graph cannot contain cycles, for example, rainfall and soil moisture are known to form a positive feedback under certain conditions (Guillod et al., 2015; Bui et al., 2023). However, while rain can affect current and future soil moisture states, soil moisture can only affect the future state of rain, not the current or past states. A graph with acyclicity and directed edges is called a Directed Acyclic Graph or DAG (Fig. 1b). DAGs are a common representation of causal relations (Pearl, 2009; Peters et al., 2017).

A DAG can also be represented as a matrix, called an Adjacency Matrix (Fig. 1c) (Peters et al., 2017). Representing DAGs as a mathematical object allows various mathematical operations to be performed on it (see Section 2.3). An adjacency matrix has its rows (or columns) named after the variables of the system and its columns (or rows) as the transpose of the former. The existence (or non-existence) of a relation between two variables is represented with a binary operator (1 and 0 or true and false)

in the corresponding cell of the matrix. To show the directionality of relations, we choose to define the adjacency matrix such that the causes reside in the rows and the effects reside in the columns (see Fig. 1c).

We note that the strength of causal relations can be represented by the coefficients of the adjacency matrix, such that the values lie between  $(-\infty,\infty)$ . However, in this paper we are only interested in the presence (and absence) of causal relations, thus we restrict the adjacency matrices to represent the same via 1's and 0's.

An interesting consequence of causality and acyclicity of DAGs is the lower triangular ordering of the coefficients of the adjacency matrix (Cunningham and Schrijver, 1998; Park and Klabjan, 2017). It can be shown that by following a simple rule (1) for reordering the rows  $r_i$  of an adjacency matrix  $A_{ij}$ , it can be converted into a lower triangular form.

$$a_{ij} = 1 \text{ if } a_i \to a_j \text{ and } i 

210

225

230

imbalance in the dataset (Chicco and Jurman, 2020).

$$MCC = \frac{TP \cdot TN - FP \cdot FN}{\sqrt{(TP + FP) \cdot (TP + FN) \cdot (TN + FP) \cdot (TN + FN)}}, \quad \in [-1, 1]$$
(3)

## (c) False positive ratio (FPR)

False positive ratio is defined by the number of False Positives identified as a proportion to True Negatives in the adjacency matrix (Powers, 2020).

$$FPR = \frac{FP}{FP + TN}, \quad \in [0, 1] \tag{4}$$

## 2.4 Synthetic model and data

We surveyed various model outputs with the following criteria in mind: a) all data generated by the model are available for use, b) all model forcing variables are available, and c) all the time-series are available at the same resolution at which they were generated or used. With these criteria, we surveyed various models (Gosling et al., 2023; Schellekens et al., 2017) and selected the Global Land Data Assimilation model Version 2.0 (GLDAS) outputs (Li et al., 2018). GLDAS primarily models the natural processes of land surface and sub-surface, with no representation of human activities like irrigation, water resources management practices like dam and canal operations. Other models such as WaterGap, PCRGLOB-WB, H08 etc, simulate such processes, however, their publicly available datasets, did not meet our above criteria.

The GLDAS dataset is a family of outputs from three Land Surface Models, Catchment Land Surface Model (Koster et al., 2000; Ducharne et al., 2000) (CLSM), NOAH-Land Surface Model and the Variable Infiltration Capacity model. We choose the output from the CLSM model. CLSM is based on the Mosaic Land Surface Model (Koster and Suarez, 1992) and adopts its energy and canopy interception routines. The model does not have vertical layers and it adopts the TOPMODEL (Beven and Kirkby, 1979) framework to simulate sub-surface moisture, defined as the average amount of water required to saturate the catchment. The vertical distribution of soil moisture profile is derived from relations explained in Clapp and Hornberger (1978). Snow is represented with a three-layer snow model described in Lynch-Stieglitz (1994).

To create an adjacency matrix from the generating equations of CLSM, we did a literature review of the model structure and equations (Appendix A), and created the True CLSM adjacency matrix (Fig. 2a), following the definitions of a DAG and its corresponding adjacency matrix (explained in Section 2.2). While a majority of the variables are generated with contemporaneous states, some variables, like storage terms (surface soil moisture storage, root-zone soil moisture, etc.), are dependent on their previous states. These are represented with lagged relations (Fig. 2a). The True CLSM adjacency matrix acts as a reference truth for our analysis, representing the cause-effect relations in the generating equations. We extracted data from eight different Köppen-Geiger Climate zones (Figures 2b-f) to understand the performance of these methods in different climates. More details regarding the forcing, simulated variables and simulation period are described in Appendix A.

**Figure 2.** a) The True adjacency matrix representing the causal relationships between the simulated and forcing variables of CLSM-F2.5 model, run under the GLDAS-2.0 dataset. The matrix is created after literature review of the model architecture and generating equations, and using the definition of adjacency matrix adopted in Section 2.2. There are 82 true positives and 1376 true negatives in the matrix. b) to f) shows the basins and the grids within, where the time-series data of simulated and forcing variables, were selected for the analysis. We extracted data from nine grid points from six major river basin of the world across eight distinct Köppen-Geiger Climate Classes. Though CLSM does not use a river routing scheme, we overlaid the HydroSHEDS (Lehner et al., 2008) river network to avoid choosing a grid over a stream, and manually selected points without any preference. g) The legend showing the various Köppen-Geiger Climate Classes.

| CD method | Modelling                                                                | Free parameters                                       |  |
|-----------|--------------------------------------------------------------------------|-------------------------------------------------------|--|
| PCC       | Finds statistically significant co-variance of variables                 | -Significance threshold                               |  |
| TCDF      | Uses Convolutional Neural Networks with attention mechanism to find      | - Hidden layers - kernel size - number of             |  |
|           | causal parents of each variable in time-series.                          | epochs - dilation coefficient - significance          |  |
|           |                                                                          | - learning rate                                       |  |
| VARLiNGAM | Fits a SVAR model in two steps. First step uses classic VAR modelling    | - Maximum lag to model                                |  |
|           | of lagged causal relations, second step uses ICA to find causal ordering |                                                       |  |
|           | of contemporaneous causal links                                          |                                                       |  |
| PCMCI+    | Explicitly finds a DAG using conditional independence tests in two       | - Significance threshold, $\alpha_{PC}$ - Maximum     |  |
|           | steps. First step uses PC-algorithm to find skeleton of causally linked  | and minimum lag to model - Conditional                |  |
|           | variables. Second step uses MCI to eliminate and direct edges in         | independence test                                     |  |
|           | skeleton.                                                                |                                                       |  |
| DYNOTEARS | Fits a SVAR model in one step. Uses continuous optimization to           | - Sparsity penalty terms, $\lambda_w$ & $\lambda_a$ - |  |
|           | reduce error of SVAR model and an acyclicity constraint                  | Maximum cyclicity allowed, $h(\mathbf{W})$            |  |
|           |                                                                          | - Coefficient threshold, $\mathbf{W}_{threshold}$ -   |  |
|           |                                                                          | Maximum lag to model                                  |  |

**Table 1.** Table summarising PCC and Causal Discovery algorithms considered for evaluation.

## 2.5 Methods: Causal Discovery algorithms

Table 1 summarizes the assumptions, modelling framework and free parameters of the various CD methods evaluated here. As mentioned above, each of these methods can be applied to a multivariate time-series dataset to unravel drivers of variables with multiple confounding, self causation, and contemporaneous and lagged causal relations. These methods adopt theoretically different modelling frameworks, for example, TCDF uses traditional Neural Networks to model time series datasets and uses GC to interpret the attention scores to unravel causal relations in the data. In turn, both VARLiNGAM and DYNOTEARS use traditional Structural Vector Auto-regressive model (SVAR) modelling to find the causal relations in the data. However, they implement different strategies to find the coefficients of the SVAR model. PCMCI+, on the other hand, uses a host of conditional independence testing to find causal drivers of a variable.

For the section below, we consider having time-series data for 'd' variables in  $\mathbf{X} = \{\mathbf{x}_t^k\}_{t \in \{0,1,\dots,T\}}$  for  $\{\mathbf{x}_t^k\} \in \mathbb{R}^d$ .

## 2.5.1 Score-based structure learning: DYNOTEARS

Introduced by Pamfil et al. (2020), DYNOTEARS seeks to find causal relations in time-series data by combining classic SVAR modelling with the acyclic property of DAGs. It models the relationships among the variables with an SVAR model and estimates its coefficients by minimizing a loss function. To ensure these coefficients represent only causal relations, DYNOTEARS considers the coefficient matrix as an adjacency matrix. Since an adjacency matrix has to be acyclic by definition, it exploits this by introducing a new term in the loss function. This new loss term represents the cyclicity of

the adjacency matrix. Thus, by simultaneously minimizing the loss of fit of the SVAR model and ensuring its acyclicity, DYNOTEARS models the relations in the data and ensures the relations are strictly causal.

Specifically, it finds the coefficients of an SVAR model with p lags:

$$x_t^k = x_t^k \mathbf{W} + x_{t-1}^k \mathbf{A}_1 + \dots + x_{t-n}^k \mathbf{A}_p, \text{ for } t \in (p, \dots, T) \text{ and for all } k \in (1, \dots, d)$$

$$(5)$$

where  $\mathbf{W}$  is a  $d \times d$  matrix containing the coefficients that capture contemporaneous relations among the variables. Thus it is equivalent to an adjacency matrix with only contemporaneous rows and columns  $(x_t^1, x_t^2, \cdots, x_t^d)$ . Similarly, the matrices  $\mathbf{A}_1, \cdots, \mathbf{A}_p$  contain the coefficients that reflect the lagged relationships between variables. Thus, it is equivalent to an adjacency matrix which represents lagged relations, hence it has both contemporaneous and lagged variables in its rows and columns but entries only in the lagged variables rows.

Further Eq. (5) can be rewritten as Eq. (6) such that  $\mathbf{X}$  is an  $n \times d$  matrix with each row containing  $x_t^k$ , while  $\mathbf{X}_{t-1}, \ldots$  are its lagged versions. This can be further compacted such that all lagged relations are represented by  $\mathbf{A} (= [A_1, \cdots, A_p])$  and contemporaneous relations by  $\mathbf{W}$  (Eq. (7)). Note that since  $\mathbf{A}$  contains only lagged relations, it only connects earlier time-steps to later ones and is inherently acyclic due to time ordering (Fig. 1b).

$$\mathbf{X}_t = \mathbf{X}_t \mathbf{W} + \mathbf{X}_{t-1} \mathbf{A}^1 + \dots + \mathbf{X}_{t-p} \mathbf{A}^p$$
 (6)

$$\mathbf{X}_t = \mathbf{X}_t \mathbf{W} + \mathbf{X}^- \mathbf{A} \tag{7}$$

DYNOTEARS estimates the coefficients of the SVAR model,  $\mathbf{W}$  and  $\mathbf{A}$ , using continuous optimization to reduce the error of fit. The loss function  $F(\mathbf{W}, \mathbf{A})$ , contains four terms (Eq. (9)). The first term is the sum of square of errors ( $\ell_2$  norm) to reduce the error of fit. Next, since the causal relations in real world data are expected to be sparse, i.e. only a few variables affect a particular variable, thus many coefficients in  $\mathbf{W}$  and  $\mathbf{A}$  are expected to be zeros. To encourage this sparsity, a penalty term is added to reduce the number of non-zero coefficients. This penalty is based on the  $\ell_1$  norm - the sum of the absolute coefficients of a matrix. This penalty term is added for both the matrices and weighted with a tuning parameter to control the degree of sparsity:  $\lambda_{\mathbf{W}} \|\mathbf{W}\|_1$  and  $\lambda_{\mathbf{A}} \|\mathbf{A}\|_1$ . Finally, to represent causal relations, the matrices  $\mathbf{W}$  and  $\mathbf{A}$  must be acyclic. As discussed earlier,  $\mathbf{A}$  is inherently acyclic. To enforce acyclicity of  $\mathbf{W}$ , a term is introduced in the loss function to penalize cyclicity in  $\mathbf{W}$ . Here the cyclicity of a matrix  $\mathbf{W}$  is expressed as a mathematical function as:

$$h(\mathbf{W}) = \operatorname{tr}\left(\exp^{\mathbf{W} \circ \mathbf{W}}\right) - d \tag{8}$$

where:

- tr is the *trace* of a matrix (the sum of its diagonal entries),
- ○ denotes the *Hadamard product* (element-wise matrix multiplication),
- and d is the number of variables.

This function equals zero if and only if **W** is acyclic. Intuitively it provides a mathematical formulation of cyclicity, as a continuous function, which can be minimized by an optimization scheme. As suggested by Zheng et al. (2018), the equality in Eq. (8) can be solved using the augmented Lagrangian method. The resulting loss function takes the form as:

$$F(\mathbf{W}, \mathbf{A}) = \frac{1}{2d(T+1-p)} \|\mathbf{X} - \mathbf{X}\mathbf{W} - \mathbf{X}^{-}\mathbf{A}\|_{F}^{2} + \lambda_{\mathbf{W}} \|\mathbf{W}\|_{1} + \lambda_{\mathbf{A}} \|\mathbf{A}\|_{1} + \alpha h(\mathbf{W}) + \frac{\rho h(\mathbf{W})^{2}}{2}, \tag{9}$$

which can be solved using standard optimization solvers like L-BFGS-B (Limited-memory Broyden, Fletcher, Goldfarb, and Shannon optimization method with bound constraints, Byrd et al. (1995)). Finally, since the causal relations are represented by coefficients of the SVAR model, some coefficients can be very small. To ignore such coefficients, the algorithm allows a user defined threshold,  $\mathbf{W}_{threshold}$ , so that only coefficients greater than this threshold represent a causal relation. Thus, DYNOTEARS offers five free parameters to control the algorithm. The two sparsity penalty terms  $\lambda_w$ ,  $\lambda_a$ , the maximum cyclicity allowed  $h(\mathbf{W})$ , the threshold of SVAR coefficients  $\mathbf{W}_{threshold}$ , and the maximum lag to search for.

#### 2.5.2 Vector Auto-regressive modelling using Non-Gaussian noise: VARLiNGAM

Similar to DYNOTEARS, VARLiNGAM, introduced by Hyvärinen et al. (2008), seeks to model the causal relations in time series data with an SVAR model. To find the model coefficients, it uses a classic least squares solution and exploits the lower triangular ordering of the adjacency matrix. Similar to DYNOTEARS, it considers the coefficients matrix as composed of a contemporaneous adjacency matrix and a lagged adjacency matrix. It starts by calculating an initial estimate of the lagged adjacency matrix, this captures the lagged relations in the data. The estimated lagged effects are then subtracted from the original data to get the residuals. Now these residuals are assumed to contain only contemporaneous relations. To find the contemporaneous causal relations in the residuals, it searches for an ordering of the variables such that the resulting matrix is lower triangular, thus representing a DAG. Finally, it uses the contemporaneous adjacency matrix to get the final estimate of the lagged adjacency matrix. Specifically it seeks to model the data with an SVAR model of p lags as:

$$\mathbf{X}_t = \sum_{\tau=0}^p \mathbf{B}_k \mathbf{X}_{t-k} + \hat{\boldsymbol{\eta}}_t, \tag{10}$$

where  $\mathbf{X}_t$  is a  $d \times n$  matrix containing the time series data for all d variables.  $\mathbf{B}_k$  is a  $d \times d$  matrix of the causal relations at lag k.  $\hat{\boldsymbol{\eta}}_t$  is a vector of errors obtained from model inaccuracy. To estimate the coefficients of  $\mathbf{B}_k$ , it breaks down the matrix into a contemporaneous matrix  $\mathbf{B}_0$ , which contains the instantaneous relations. While  $\mathbf{B}_k$  for  $k = 1, \ldots, p$ , contains the lagged relations. It begins by calculating an initial estimate,  $\hat{\mathbf{M}}_k$ , of the lagged relations  $\mathbf{B}_{k, k>0}$ , using an ordinary least squares solution. Then it removes the effect of lagged relations from the data to get the residuals as:

$$\hat{\mathbf{u}}_t = \mathbf{X}_t - \sum_{k=1}^p \hat{\mathbf{M}}_k \mathbf{X}_{t-k}, \tag{11}$$

these residuals are assumed to contain only contemporaneous relations. To unravel these relationships it uses LiNGAM analysis.

The Linear Non-Gaussian Structural Equation Model or LinGAM analysis was introduced by Shimizu et al. (2006). LinGAM allows modelling of causal relations in a vector regressive model (i.e. a SVAR model with no time delays). To

330

335

find the coefficients matrix, it searches for an ordering in the columns of the matrix such that the resulting matrix is lower triangular and hence equivalent to a DAG representing causal relations. Note that for a small number of variables this can be done by following the steps in Eq. (1). However, for a large number of variables this becomes computationally expensive. LiNGAM finds this ordering by posing the problem as a classic Independent Component Analysis (ICA) problem (Hyvärinen et al., 2001). Consider Eq. (11) written as:

$$\mathbf{315} \quad \hat{\mathbf{u}} = \mathbf{B}_0 \hat{\mathbf{u}} + \mathbf{e} \tag{12}$$

$$\hat{\mathbf{u}} = \mathbf{Q}\mathbf{e}$$
, where  $\mathbf{Q} = (\mathbf{I} - \mathbf{B}_0)^{-1} = \mathbf{W}^{-1}$ , (13)

Here we drop the time subscript since all relations are at the same time-step. The aim is to find a permutation of the matrix  $\mathbf{W}$  such that it has ones on its diagonals. So that  $\mathbf{W} = \mathbf{I} - \mathbf{B}$  yields a matrix with zeros on its diagonals, which is a requirement of an adjacency matrix representing a DAG. To do this, it decomposes the  $\mathbf{W}$  using ICA into  $\mathbf{W} = \mathbf{PDW}$  where  $\mathbf{D}$  is a diagonal matrix and  $\mathbf{P}$  is the particular permutation matrix which yields ones on the diagonals of  $\mathbf{DW}$ . Thus we obtain the estimate for the contemporaneous matrix  $\mathbf{B}$ , which is used to update the estimates of the lagged adjacencies using Eq. (14).

$$\hat{\mathbf{B}}_k = (\mathbf{I} - \mathbf{B}_0)\hat{\mathbf{M}}_k \quad \text{for } k = 1, \dots, p$$

Thus VARLiNGAM has one free parameter, the maximum lag parameter, to control the application of the algorithm.

## 2.5.3 Constraint-based causal discovery: PCMCI+

The constraint based PCMCI+ algorithm (Runge, 2022) uses conditional independence (CI) tests to find causal parents (drivers) of variables in multi-variate time-series data. It achieves this in two steps, *i*) skeleton identification phase using a modified form of PC-alg (PC<sub>1</sub>), to model lagged relations, and *ii*) full skeleton phase using Momentary Conditional Independence tests, to discover contemporaneous relations.

In the first phase PCMCI+ creates a skeleton, i.e. an undirected graph, of all plausible lagged relations. Thus a graph  $\mathcal{G}$  is initialized with all possible edges between pairs of contemporaneous and lagged variables. Then to remove the non-causal relations (edges) it uses CI testing. It uses the PC<sub>1</sub> algorithm to reduce the number of CI tests required. Thus it ends with a partially directed graph representing lagged causal relations.

The second phase is designed to identify contemporaneous and self causation. It begins by re-initializing the graph  $\mathcal{G}$  obtained at the end of the first phase. Once again CI tests are used to remove non-causal edges. Here it uses Momentary Conditional Independence tests, which unlike  $PC_1$ , also considers contemporaneous and self causation (Runge et al., 2019a). Additionally, collider orientation and rule orientation phase are used to orient any un-oriented contemporaneous or ambiguous links. Thus, it ends with a DAG likely representing the underlying causal relations in the data. We briefly explain the algorithm below.

The skeleton identification phase begins by creating a skeleton of all possible lagged relations. Here it starts with a fully connected undirected graph,  $\mathcal{G}$ , with edges between all pairs of contemporaneous variables and their lagged versions (up-to the maximum anticipated lag, p). Such that for a particular variable  $X_t^j$ , all possible (lagged) parents are considered. Let the set of plausible parents for  $X_t^j$  be  $\hat{Pa}(X_t^j) = \mathbf{X}_t^- \setminus X_t^j$ , where  $\mathbf{X}_t^- = \{X_{t-1}^k, X_{t-2}^k, X_{t-p}^k\} \ \forall \ k \in (1, 2, \dots, d)$ .

Now, it tests for CI between  $X_t^j$  and one of its plausible parents from  $\hat{Pa}(X_t^j)$ , say  $X_{t-\tau}^i$ , by conditioning them against the remaining parents, if the hypothesis, Eq. (15), is not rejected at a desired significance level  $\alpha_{PC}$ , then the variable is removed from the set of plausible parents  $\hat{Pa}(X_t^j)$  (consequently the edge is removed from  $\mathcal{G}$ ).

$$X_t^j \perp \perp X_{t-\tau}^i \mid S$$
 where  $S \subseteq \hat{Pa}(X_t^j) \setminus \{X_{t-\tau}^i\}$  and  $|S| = \tau$  for  $\tau = 0, 1, \dots, p$  (15)

For a given size of parent set, say L, a high number of combinations for the conditioning set S can be generated  $(2^L)$ , which is also the problem faced by TE (Runge et al., 2012). This makes the task of pruning edges with CI tests computationally expensive, while a large conditioning set reduces the strength of the CI tests (Runge et al., 2019b). As mentioned earlier, PCMCI+ uses a modified form of the PC-alg, PC<sub>1</sub>, to reduce the number of CI tests required. The algorithm starts with the smallest possible conditioning set  $(\tau = 0)$ , where  $|S| = \tau$  and iteratively increases its size until the parents in  $Pa(X_t^j)$  are exhausted in the conditioning set, *i.e.* all possible parents of  $X_t^j$  form the conditioning set  $S(S = \hat{Pa}(X_t^j))$ . Thus by prioritizing smaller conditioning sets in the CI tests, it reduces the size of  $Pa(X_t^j)$  and also preserves the strength of CI tests with smaller size of the conditioning set  $S(S = R^j)$  (Runge et al., 2012).

Now, within each p-th iteration, the conditioning set can have different variables and their combinations as the conditioning set. This can quickly lead to an extremely high number of CI tests to be performed. For example, if  $|Pa(X_t^j)| = 8$  and  $\tau = 3$ , the number of CI tests performed would be  ${}^8\mathrm{C}_3$ . To avoid this, the algorithm tests only against the strongest p combinations of the conditional set. Therefore for  $\tau = 0$ , the conditioning set is empty and the CI test is equivalent to a correlation analysis. The algorithm sorts the parent set  $\hat{Pa}(X_t^j)$  according to the strength of correlation in the previous step. For  $\tau = 1$ , the CI test is equivalent to a partial correlation analysis, and so on. Where it tests for CI using only the first (strongest correlated) variable from  $\hat{Pa}(X_t^j)$  in the conditioning set S.

To deal with auto-correlation in the time series and find the contemporaneous links, authors use the Momentary Conditional Information test (MCI) (Runge, 2022). The main difference between PC<sub>1</sub> CI test and MCI test is that the latter considers the causal parents of the variables undergoing the CI test in the conditioning set itself (Eq. 16) (Runge et al., 2019a). Thus in the second step, the graph  $\mathcal{G}$  is re-initialised by adding all the contemporaneous links  $\hat{A}(X_t^j)$  possible. Now, similar to PC<sub>1</sub>, pruning and orientation of edges follows using the MCI test (Eq. 17).

$$X_t^j \perp \!\!\! \perp X_{t-\tau}^i \mid P(X_t^j) \setminus \{X_{t-\tau}^i\}, P(X_{t-\tau}^i)$$

$$\tag{16}$$

$$X_{t}^{j} \perp \!\!\! \perp X_{t-\tau}^{i} \mid S, \ B_{t}^{-}(X_{t}^{j}) \setminus \{X_{t-\tau}^{i}\}, \ B_{t-\tau}^{-}(X_{t-\tau}^{i}), \ \text{where} \ S \subseteq \hat{A}(X_{t}^{j}) \setminus \{X_{t-\tau}^{i}\} \ \text{and} \ |S| = \tau \ \text{ for } \tau = 0, 1, \dots, p \tag{17}$$

 $B_t^-(X_t^j)$  and  $B_{t-\tau}^-(X_{t-\tau}^i)$  are the causal parents of  $X_t^j$  and  $X_{t-\tau}^i$  respectively, identified at the end of PC<sub>1</sub>. S is a subset of contemporaneous adjacencies of  $X_t^j$ . Finally, any undirected contemporaneous edges in  $\mathcal G$  are oriented using PC-alg's orientation rules (Spirtes and Glymour, 1991). Amongst the CD methods discussed here, PCMCI+ offers the highest flexibility to adapt the algorithm for discovery in linear and non-linear time-series datasets. Thus it has several free parameters, starting with the significance level of the CI tests in both the PC<sub>1</sub> and MCI tests ( $\alpha_{PC}$ ), second it allows the use of any linear or non-linear (user defined) test for independence in both the stages (PC<sub>1</sub> and MCI). Third, the maximum and minimum lag up-to which the lagged relations are anticipated.






#### 2.5.4 Granger causality based: Temporal Causal Discovery Framework

Introduced by Nauta et al. (2019), Temporal Causal Discovery Framework or TCDF identifies causal drivers of variables by combining deep neural network based modelling with a GC-inspired interpretation of model weights. TCDF can be divided into two broad steps, the first step involves identifying the potential causes of each variable by training deep neural networks. The second step uses the structure of the trained model to determine when a discovered causal driver has its effect (lagged and/or instantaneous).

The first step forms the major analysis of TCDF, it can be broadly divided into three parts, where for each variable in the data it begins by a) training a deep neural network model—a Convolutional Neural Network (CNN) to predict it, b) it uses the *attention* of the trained model to identify the potential causes. The attention mechanism of a CNN model helps it to focus on certain variables when predicting a target variable, and c) to verify the potential causes as true causes, it conducts a feature importance step by randomly permuting the values of a potential cause and predicting the target variable. Thus the first step ends by identifying the (likely) true causes of a variable and its corresponding trained CNN model.

In the second step, TCDF determines the temporal order of relation between the identified causes and the target variable. To do this TCDF simply interprets the kernel weights of the trained CNN model. Where TCDF traverses from the output layer (target variable) to the input layer (discovered cause), taking the path with the highest kernel weights. The position where it meets the input layer is decided as the order of the lagged relation. Both steps are repeated for the remaining variables in the system to identify all causal relations in the system. We describe the algorithm below, for a detailed description we suggest reading Nauta et al. (2019).

TCDF begins by training an independent CNN model for each variable. Thus, for each (target) variable  $\mathbf{X}^{\mathbf{j}}$  (for  $j=1,\ldots,d$ ), it uses an independent CNN,  $N_j$ , to model the patterns in its time-series. This network uses all the other variables and their lags and past values of  $\mathbf{X}^j$ . Thus network  $N_j$  is responsible for modelling  $\mathbf{X}^j$  and its causes. Inside  $N_j$ , channels  $n_i$  (for  $i=1,\ldots,j,\ldots,d$ ), exist, which are responsible for modelling the relation from a variable  $\mathbf{X}^i$  to  $\mathbf{X}^j$ . Note that  $n_j$  models self causation. Next, to identify potential causes of  $\mathbf{X}^j$  it uses a GC-inspired approach. TCDF considers a variable  $\mathbf{X}^i$  as a potential cause if it improves the prediction in  $N_j$  by reducing the model loss. To identify which variables the  $N_j$  considers important, TCDF uses the attention mechanism (attention vector  $\mathbf{a_j}$  Eq. 18) associated with it. These attention vectors are  $1 \times N$  and their coefficients tell how much attention was paid by  $N_j$ , to a certain time-series when predicting  $\mathbf{X}^j$ .

$$\mathbf{a_j} = [a_{1,j}, \dots, a_{j,j}, \dots, a_{N,j}]$$
 (18)

where  $a_{i,j} \in \mathbf{a_j}$  is called an attention score, which represents the attention given to  $\mathbf{X}^i$  by  $N_j$  when predicting  $\mathbf{X}^j$ .

The more attention a variable receives, the more likely it is to be considered a causal influence. Since attention scores take continuous values between [0,1], TCDF applies a threshold to convert them into binary decisions (causal or non-causal) attention. Thus, if  $a_{i,j}$  exceeds a certain *threshold*,  $\mathbf{X}^i$  is considered a potential cause  $(P_j)$  of the *j-th* variable. Finally, to verify if the identified causes are indeed true causes it uses a feature importance method called Permutation Importance Validation Method, we briefly define below.




For each potential cause  $X^i \in P_j$ , a new dataset is created by intervening into the system. This is done by randomly permuting the values of  $X^i$  to destroy their chronological ordering while keeping the values of other variables the same. The trained CNN model in the previous step is run again using the intervened data and the model loss is compared to the previous scenario where no intervention (via permutation) was done. If the loss is *significantly* higher after disturbing the values of the potential cause, it is considered to be a true cause (Nauta et al., 2019).

The final step involves determining the temporal order of causal relations between the identified causes in  $P_j$  and  $\mathbf{X}^j$ . For this TCDF simply uses the kernel of the trained CNN model. The kernel is a convolution operator between  $\mathbf{X}^i$  (the input layer) and its effect  $\mathbf{X}^j$  (the output layer). Specifically, the kernel is a weight matrix of size  $N \times K$ , where K is the kernel size. These K weights represent the influence of respective delays on the output. Thus by following the path from the  $K^j$  to  $K^j$  via the highest weights (coefficients) in the kernel matrix, the position in  $K^j$  can be identified which has the maximum influence on  $K^j$ . This position is considered to be the lag in the cause-effect relation between  $K^i \to K^j$ . As mentioned earlier, this entire process is repeated for all the variables in the data to identify causal relations in the system.

As with any deep learning method, TCDF has several hyper-parameters. It requires tuning of a number of hidden layers, kernel size, number of epochs, learning rate, dilation coefficient and significance (Nauta et al., 2019; Assaad et al., 2022).

## 2.5.5 Other methods for Causal Discovery

We note that other distinct methods of discovering causal relations do exist for example, based on difference equations, which represents all causal relations by means of difference equations driving changes in the system (Voortman et al., 2010). Further, based on non-linear state space reconstruction–CCM (Sugihara et al., 2012), etc. For a comprehensive review we suggest reading Assaad et al. (2022); Gong et al. (2024); Ali et al. (2024). As mentioned earlier, CCM has been successfully applied to discover causal relations in hydrological systems. However, we did not select it for evaluation due to two major issues. First, being a bi-variate method, it allows to determine causality only between a pair of variables, thus it is highly susceptible to identify incorrect causal relations in multi-variate system as discussed earlier (Ombadi et al., 2020). Second, more importantly it assumes a deterministic system in order to create the high dimensional manifold which represents the dynamical and thus consistent (causal) relations in the data (Sugihara et al., 2012). In hydrology, observational and process noise are typical in observations. As shown by Ombadi et al. (2020), applying CCM in such systems can lead to reduced power of detecting causal relations. Despite these, it remains a strong candidate for discovering causal relations, when the assumptions are satisfied.

The choice of free parameters for the four CD methods described above was based on the suggestions from their respective papers. Details of these settings, along with those for the PCC method, are provided in Appendix B. In the next section we discuss the set of assumptions adopted by CD methods in order to discover causal relations.

# 2.5.6 Assumptions

The ability of CD methods to discern causality from correlation lies in the statistical measures used by them and the definition of dependence adopted by them – via DAGs. These rely on two sets of assumptions: one about the nature of the data and the other on the recoverability of the underlying DAG. Thus, assumptions of Gaussian distribution of variables and stationarity of


time-series are common to each method, except VARLiNGAM. Assumptions related to the recovery of the underlying DAG are (a) Causal Sufficiency, (b) Markov Assumption, and (c) Faithfulness, (Assaad et al., 2022). Below we briefly define these DAG related assumptions, while Table 2 lists the algorithms which adopt them.

| Method    | Causal Sufficiency | Markov Assumption | Faithfulness |
|-----------|--------------------|-------------------|--------------|
| PCC       |                    |                   |              |
| TCDF      |                    |                   |              |
| VARLiNGAM | ✓                  | ✓                 |              |
| PCMCI+    | ✓                  | ✓                 | ✓            |
| DYNOTEARS | ✓                  |                   |              |

Table 2. Causal discovery assumptions. An empty cell indicates the assumption is not needed.

Causal Sufficiency, requires that all the variables which are anticipated to affect the system be included in the analysis.

For example, if root zone soil moisture acts as a causal driver of surface soil moisture and transpiration, but it is unobserved, a causal analysis would wrongly yield a causal link between the latter two. Such cases of unobserved variables also result in discovery of incorrect lagged links (Runge, 2018).

**Markov** assumption implies that a DAG is supported by the conditional independencies present in it. More formally, for the joint distribution of variables in X with Graph G, the causal structure in G is supported by corresponding conditional independence tests. For example, the structure in graph G, Eq. (19), with only two links into Transpiration. The Markov assumption implies that this graph is supported by the conditional independence tests in Eq. (20).

$$G \equiv \text{Total-Precipitation}_t \rightarrow \text{Transpiration}_t \leftarrow \text{Root Zone-Soil moisture}_t$$
 (19)

$$\mathbf{X} \setminus Pa(\operatorname{Transpiration}_t) \perp \operatorname{Transpiration}_t \mid Pa(\operatorname{Transpiration}_t)$$
 (20)

where  $Pa(\text{Transpiration}_t)$  is the Causal parent set of  $\text{Transpiration}_t$  and consists of  $\text{Total-Precipitation}_t$  and  $\text{Root Zone-Soil moisture}_t$ .

In contrast to the Markov assumption, the **Faithfulness assumption** implies that all conditional independencies of various disjoint sets in **X** are represented in the graph G. Thus if we are to find the conditional independence in Eq. (20) to be true, the Faithfulness assumption necessitates it to be represented in the structure G, Eq. (19).

## 2.5.7 Time-series prediction model

To understand the effect of identifying various drivers (causal and non-causal) of a variable, we evaluated the difference in predicted surface soil moisture time-series when using drivers identified by PCC and the CD methods. In recent times causal discovery has been used in four different ways for time-series predictions. First, Yuan et al. (2022), used the difference in cross entropy amongst observed and simulated variables as a loss function in addition to the sum of square of errors, to train a deep learning model for predicting wetland methane emissions. Second, Li et al. (2022) used the adjacency matrix both, as a feature selection step and to modify the gates of their LSTM cell for soil moisture prediction. Third, Wu et al. (2025) used the adjacency

matrix to introduce a causal inference unit alongside the LSTM cell, in their spatiotemporal soil moisture estimation model. Fourth, Vázquez-Patiño et al. (2022) used causal discovery to identify robust features for spatial downscaling of precipitation. Similarly, Zou et al. (2023) used causal discovery to identify the drivers of irrigation water use, to build a prediction model.

Similarly, we use PCC and CD methods to identify the predictors of surface soil moisture in the Ganga basin. Then, we train a machine learning model, based on the sets of predictors. Using the CLSM data, we train the models from 01 January 2000 to 31 December 2003. While we evaluate their performances during the drought period from 01 January 2004 to 31 December 2005. Furthermore, we conducted a similar exercise for storm surface runoff prediction in Ganga basin and transpiration prediction in the Murray basin, and obtained similar results (Appendix C).

Since model training and evaluation is done using the CLSM data, the models will achieve a near perfect fit irrespective of the number of causal and non-causal predictors identified, or the model structure (Appendix C). This is a result of the perfect model environment, without observational or process noise in the simulated data. Thus, to understand the effect of observational noise, typically present in hydro-meteorological data and conduct this exercise in a more realistic scenario, we added randomly generated gaussian noise to the data. To ensure robustness against the distribution of randomly generated noise, we conducted Monte Carlo simulations for different noise levels (Appendix C).

## 3 Results



To evaluate the performance of the CD methods relative to PCC, we adopt two broad approaches. First, we see the performance at the macro scale by evaluating the adjacency matrices. Second, we zoom into the analysis by focusing on the drivers of surface soil moisture identified by different methods across all the grid points. Finally, to understand the consequence of finding causal and non-causal drivers in terms of applications, we use machine learning models to predict the surface soil moisture time-series. These models are trained separately with predictors identified by PCC and CD methods.

# 485 3.1 Can CD methods identify the true drivers in a complex simulated hydro-meteorological system?

The primary aim of any predictor discovery algorithm is to identify the true drivers of the target variable. To evaluate this, we consider the Recall (or True Positive Ratio, TPR) of all algorithms across the Köppen-Geiger climate classes in Fig. 3a.

Overall, across the Köppen-Geiger climate classes, PCC identifies the highest number of links present in the true adjacency matrix (Fig. 2a). While DYNOTEARS shows lower Recall than PCC, the other CD algorithms identify half or fewer causal links. The cumulative plot indicates that PCC exhibits the largest inter-quartile range (IQR). While CD methods show a narrower IQR, in the order PCMCI+ 


basin, VARLiNGAM in Danube basin and DYNOTEARS in Ganga basin. While PCMCI+ remains relatively stable across all climate types. Overall, CD methods show a relatively stable Recall across climate types.

Figure 3. Recall (or true positive ratio) (a) and Matthews Correlation Coefficient (b) for all the algorithms, across different Köppen-Geiger climate classes and in different river basins. The right most boxplots show the cumulative distributions with the median values annotated on the y axis. Note that both the top and bottom labels are common to a) and b). The legend is common to a), b). Recall is simply the ratio of true positives identified to the actual number of true positives in the reference truth,  $Recall = \frac{TP}{TP+FN}$ ,  $\in [-1,1]$ . MCC consider the class imbalance by using all four classes,  $MCC = \frac{TP \cdot TN - FP \cdot FN}{\sqrt{(TP+FP) \cdot (TP+FN) \cdot (TN+FP) \cdot (TN+FN)}}$ ,  $\in [-1,1]$ 

# 3.2 What is the overall performance, in terms of identifying causal relations and eliminating non-causal co-relations?

As mentioned in Methods, Recall does not consider the other classes of (mis)identification, such as false positives, nor the imbalance in their size. This is especially relevant to our analysis since the True adjacency Matrix is negatively imbalanced with



90% negatives (1376 negatives and 82 positives). Thus, we use Matthew's correlation coefficient (MCC) score to get a balanced understanding of performances. After considering the class imbalance, we observe a change in the relative performance of all the algorithms (Fig. 3b). We explore these differences below.

The cumulative plot indicates that PCC has the lowest MCC scores, with median MCC 0.14. While CD methods score a median MCC greater than or equal to 0.19. Although PCC has the highest Recall, it has very high false positives, resulting in a lower MCC. Among CD methods, TCDF and VARLiNGAM yield comparable median MCC values, but TCDF achieves higher MCC within the IQR. Similar to Recall results, PCMCI+ achieves the most stable MCC scores, while DYNOTEARS shows the largest IQR amongst all the CD methods. The variability of IQR among CD methods follows the order PCMCI+ 

Figure 4. Scatter plot of True Positive Ratio and False Positive Ratio in all the grids. The size of the points shows the absolute number of false positives categorically via the False Positives legend. The red dotted line represents a case where TPR=FPR, the top left 'Ideal' point denotes a perfect scenario where no False positives are detected and all true positives are identified. The angles in-set show the angle between an imaginary line from the origin to the median of each point cloud and the FPR axes  $(\arctan(\frac{TPR_{median}}{FPR_{median}}))$ .







## 515 3.3 What is the trade off between choosing a correlation-based approach and CD methods?

To better understand the balance between True Positive discovery (Recall) and False Positive discovery we plot them in Fig. 4 for each method. As seen in the plot, the cost of identifying causal links is the accumulation of false positives. Overall, all the algorithms achieve a higher TPR compared to FPR (they sit above the red dotted line, which represents TPR=FPR). Amongst the CD methods, DYNOTEARS achieves the highest TPR, but also suffers from the highest FPR. Further, CD methods show variance along the TPR axis but less variance along the FPR axis. This demonstrates their robustness towards eliminating false positives. PCC shows high variance along both axes, lacking robustness in identifying true positives and avoiding false positives. In terms of absolute number of false positives, CD methods identify less than 200 links incorrectly, whereas PCC identifies between 600 to 1000 incorrect links as true positives. Overall, across the methods, we observe a higher TPR entails a higher False Positive discovery. Since the range of TPR and FPR is significantly different across the methods, we calculated the ratio of TPR to FPR for each method. That is the angle between an imaginary line connecting the origin to the median point of each point cloud, and the FPR axes (Fig. 4). It can be clearly observed that the CD methods, compared to PCC, achieve a higher TPR gain for a unit increase in FPR and a larger deviation from the TPR=FPR line.

## 3.4 Identifying causal drivers of hydrological variables and time-series predictions.

The previous results sections reported results across all causal relationships within the CLSM model. However, to better understand what the variance in FPR and TPR means in practice, we extract all the drivers of an individual variable, surface soil moisture, identified by the algorithms across all the grids in each climate and plot them in Figures 5 and 6. Surface soil moisture is an important hydro-meteorological variable as it links the atmosphere with terrestrial hydrology (Seneviratne et al., 2010). In nature, the soil surface stores moisture from the atmosphere and provides moisture back to the atmosphere via evaporation. Active research is ongoing to understand the causality and timescales of this feedback system (Tuttle and Salvucci, 2017; Chauhan et al., 2023; Devanand et al., 2018).

In the CLSM model, surface soil moisture is modelled by combining various modelling routines. We define it explicitly in Appendix A. Essentially, it is modelled as a reservoir of moisture. The initial value of surface soil moisture is based on the catchment deficit (from full saturation), profile soil moisture. It receives input flux from above ground as excess precipitation and from below the ground as excess root zone soil moisture. While outgoing fluxes are direct evaporation from soil and infiltration into the root zone. To update its state at a time instant, the CLSM model takes the summation of these fluxes and adds (or subtracts) from the storage at the previous time-step. Thus, eight variables form the causal parents of surface soil moisture. These are (i) profile soil moisture, (ii) canopy interception and (iii) total precipitation (iv) precipitation as rain (v) total evaporation (vi) evaporation from bare soil (vii) root zone soil moisture, and (viii) surface soil moisture at the previous time step. Below we evaluate the ability of the algorithms to identify the causal parent of surface soil moisture in each grid of the different climates.

21

**Figure 5.** Panels (a-d) shows the various causal drivers of Surface soil moisture as identified by the algorithms in each grid across different climates. The variables left of the solid red line are the causal parents, of surface soil moisture, extracted from the True adjacency matrix, Fig. 2a. Whereas the variables to the right are all the remaining variables of the system and their lags. A blue coloured cell indicates the algorithm has identified a causal link to surface soil moisture from the corresponding variable (column) in the given climate grid (row). Similarly, a grey coloured cell indicates no causal link detected. **Note:** the legend in Fig. 6c is common to both, Figures 5 and 6

**Figure 6.** Panel a) same as Fig. 5 but for DYNOTEARS. Panel b) summarizes the previous panels across all climates and grids for each causal parent individually and collectively for the True and False Positives, for each algorithm. Panel c) is the legend common to both, Figures 5 and 6.

# 3.4.1 PCC



Figure 5a shows the drivers of surface soil moisture as identified by PCC across the different climate classes. PCC identifies all eight causal parents in the tropical and arid climates and only misses the direct evaporation from soil in a single grid of the temperate climate in the Ganga basin. Direct evaporation from soil is also the variable PCC is most likely to miss in the other climates as well (Fig. 6b). Overall, PCC identifies all eight causal parents in each climate at least once or more.

However, PCC also classifies a very large number of non-causal variables as drivers, resulting in large false positives. For example, variables such as latent net heat flux, downward short-wave radiation are closely related to surface soil moisture and part of the surface energy budget and are also identified as causal drivers in a majority of the grids. Similarly, water budget variables like storm surface runoff, groundwater storage, are also classified as causal drivers. Though such variables may have a direct impact on surface soil moisture in the natural environment, these variables are absent in the generating equations of the CLSM model. Hence do not form the causal parents. Thus, PCC showed a systemic error by identifying many variables as drivers, consistently across different climate classed.

A striking feature of CD methods is their ability to correctly eliminate lagged variables as false positives. Whereas PCC classifies each lagged relation of the causal parent as a causal driver, the CD methods eliminated these demonstrating their ability to handle auto-correlated variables.

Interestingly, PCC was able to identify canopy surface water as a causal driver. Since the canopy water acts as a reservoir above the surface and allows rainfall to reach the surface only if it is full to its capacity, it adds some non-linearity to the generation of surface soil moisture. Among CD methods, only VARLiNGAM and DYNOTEARS were able to identify this driver in at least half of the grids (Fig. 6b).

#### 565 **3.4.2 TCDF**


TCDF misses the canopy surface water and evapotranspiration as a causal driver in most of the climates (Fig. 5b). While it also struggles to identify evaporation from bare soil and profile soil moisture as well. However, it shows consistency in identifying rain precipitation rate, root zone soil moisture and the lagged self-causation of surface soil moisture.

Further, it shows a systematic error by falsely identifying the latent net heat flux as a possible driver in the Arid climates and lagged relation from baseflow in Tropical climates incorrectly. Apart from these, no other variable is consistently misidentified. Overall, TCDF achieves the fewest false positives across all climates (false positives = 0.03), making it the most conservative in terms of spurious detection.

## 3.4.3 VARLINGAM

VARLiNGAM shows contrasting results in terms of identifying the causal parents. It was able to identify six out of the eight causal drivers correctly in at least half of the grids (Fig. 6b). While it failed to identify evaporation from bare soil and evapotranspiration in most of the grids. Interestingly, it was able to correctly identify the non-linear link from canopy surface water in numerous grids across all climate classes, whereas TCDF and PCMCI+ missed this link.

In terms of false positives, VARLiNGAM also showed systemic bias, by incorrectly identifying canopy water evaporation as a causal driver. Further it falsely identified canopy water evaporation in most of the grids. Similarly, it failed to eliminate terrestrial water storage in all climates except arid, and ground heat flux and specific humidity in arid climates.

Interestingly, it attributed a lagged causal link between surface soil moisture and root zone soil moisture instead of the true contemporaneous causality. Overall, VARLiNGAM identified a higher number of causal drivers while maintaining a lower false positive count (false positives = 0.14), though it showed systemic error against some variables.

### 3.4.4 PCMCI+


The PCMCI+ method, similar to TCDF, fails to identify the canopy surface water and evaporation from bare soil. However, it identifies the remaining six drivers consistently in each climate class. PCMCI+ also shows a systemic error by falsely identifying canopy water evaporation, storm surface runoff and terrestrial water storage as causal drivers. Compared to VARLiNGAM and DYNOTEARS, PCMCI+ has a very sparse false positive detection, similar to TCDF (false positives = 0.07).




#### 3.4.5 DYNOTEARS

DYNOTEARS, in strong contrast to the other CD methods, both in terms of true positive and false positives, identifies seven causal drivers in at least half of the grid points, across all the climates. Interestingly, it could identify the lagged self causation of surface soil moisture in less than half the grids. Compared to other CD methods and PCC, this is very low, given the strong autocorrelation usually present in storage variables. While it failed to identify plant canopy surface water, evaporation from bare soil and evapotranspiration in the tropical climates.

It also showed systemic error, failing to eliminate net long wave radiation flux, average surface skin temperature, baseflow, terrestrial water storage and wind speed. Interestingly, it identified the fewest *lagged* variables as causal drivers. Overall, DYNOTEARS identified more causal drivers of surface soil moisture than the other CD methods while only identifying a few more false positives (= 0.19).

#### 3.4.6 Summary of identifying causal drivers

Overall, PCC identified nearly all of the causal drivers, across all the climates. However, it also identified a large number of false positives. Comparatively, CD methods were able to reduce the false positive count by orders of magnitude, while showing much fewer systemic errors. However, they showed less consistency in identifying all the causal drivers. To understand the effect of missing a few causal drivers and identifying non-causal ones, we compare the difference by creating prediction models. In the next section, we train machine learning models based on the predictors of surface soil moisture identified by PCC and CD methods, and evaluate their performance.

## 3.5 Predicting time-series using causal knowledge

Below we discuss surface soil moisture predictions under a noise level of 0.5 standard deviation, using a feedforward neural network model.

In the training period, PCC identified 47 drivers of surface soil moisture, of these 8 were the causal drivers discussed earlier, while 39 were non-causal variables. Similarly, TCDF, VARLiNGAM, PCMCI+ and DYNOTEARS identified 4 causal (4 non-causal), 3 causal (12 non-causal), 6 causal (3 non-causal) and 4 causal (5 non-causal) drivers, respectively. Figures 7a and 7c show the performance and error metrics respectively during this period. The PCC-based model achieves the highest accuracy relative to its training data, with median  $R^2$ , NSE > 0.8. However, it suffers a sharp decline in performance and gain in error when predicting out of sample during drought conditions. This may be a result of the high number of false positives identified as causal drivers. In contrast, the CD-based models obtain satisfactory performance metrics during the training period with median  $R^2$ , NSE > 0.75, while they show a smaller drop in performance testing out of sample during drought conditions, with median  $\Delta R^2 \approx -0.15$  and median  $\Delta NSE 

Figure 7. Performance and error metrics of the machine learning models created for surface soil moisture prediction. Panels a) and c) show the performance and error metrics during the training period. While panels b) and d) show the difference in performance and error metrics between the testing and training, eg:  $\Delta R^2 = R_{\text{testing}}^2 - R_{\text{training}}^2$ . Panel e) shows the predicted and actual time-series in the testing period, based on PCC and CD-based models. For each method, the plot shows the mean prediction of the 100 Monte Carlo simulations, while the shading shows the minimum and maximum range. (A  $\Delta(\cdot)$  < 0 for performance metrics means a drop in performance during the testing period compared to the training period. While a  $\Delta(\cdot)$  > 0 for error metrics means a drop in performance during the testing period compared to the training period.)

Overall, PCC-based models identify a large number of predictors and perform better in the training period, but suffered larger performance losses when tested under changing conditions. CD-based models obtain a parsimonious predictor set. This leads to smaller variance in performance during the testing period. More significantly, CD-based models show smaller drop in performance compared to PCC based models, when tested during changing conditions like droughts.

#### 4 Discussion






Below we discuss the capabilities of the different algorithms, discuss some caveats to applying Causal Discovery in general and in particular for Hydrology. We close the section with some perspectives on implementing CD methods and discuss limitations of our work.

As discussed in the introduction, Hydro-meteorological systems have highly interconnected variables with strong feedback mechanism and closely related processes. This introduces numerous contemporaneous and lagged correlations in the system. Thus identifying the true causal drivers of a process becomes a challenging task. Thus, a multivariate and cause-effect driven approach is needed to unravel the causal causal drivers of processes.

By applying multivariate and stochastic framework based causal discovery algorithms, we were able to recover about half of the causal drivers in the system. Further, by considering the possibility of lagged relations in the data CD methods were able to eliminate large numbers of contemporaneous and lagged spurious correlations. This provided a parsimonious set of drivers for different variables, which can potentially lead to better process understanding and identifying the causes of change like streamflow change, droughts etc. However, it also meant CD methods could not detect many causal links in the system.

While Ombadi et al. (2020) applied their CD methods on a simple model forced with stochastically generated rainfall, we evaluated our CD methods on a large, complex model forced with realistic forcings, and evaluated the results across diverse climates and basins of the world. We observed the consistency of CD methods to identify cause-effect relations across these regions. This suggests their viability to be applied in diverse hydrological systems.

Further by focusing on the drivers of surface soil moisture, we found PCC systematically identified large number of correlations with many closely related variables of the energy and water budget, of which a small subset were the causal drivers. This misidentification was consistent as a systemic error, across the different climate regions considered. Whereas CD methods showed such systemic error against far fewer variables, however they missed identifying certain causal drivers of surface soil moisture.

To understand the effect of finding causal and non-causal drivers of a variable in terms of time-series prediction, we applied machine learning models to predict surface soil moisture time-series in drought periods. In this regard, we saw (i) CD-based models are more parsimonious compared to PCC-based models, (ii) Both PCC and CD-based models perform good under the training period, (iii) importantly, CD-based models suffer a smaller drop in performance during the evaluation period. This highlights the importance of identifying causal drivers of a process, both for process understanding and predictions, especially under changing conditions like drought and possibly climate change.

## 4.1 Method specific outcomes

#### 4.1.1 PCC







As we expected, PCC found a high number of co-varying variable pairs. Choosing low threshold of correlation allowed it to discover most of the causal links (median Recall = 0.83). However, the definition of PCC lacks a causal interpretation, thus the variables identified can be called *predictors* and not causal drivers. This is the reason why any statistical test like PCC, Spearman's rho, Kendall's  $\tau$ , or measures of Information Flow like Transfer Entropy, Mutual Information etc., are called Variable Selection or Predictor importance step. Overall, and in particular for drivers of surface soil moisture, PCC identifies very high false positives. This creates an illusion of model complexity by including redundant variables and their lags, which are statistically significant but causally unrelated. This makes understanding the process harder, while it necessitates the need of many variables for creating a prediction model that inherently has high computation cost in terms of time and memory.

#### 4.1.2 TCDF

TCDF identified the fewest causal links across all the climates, however it was able to keep false positives to the lowest. As mentioned earlier, for each variable in the system, it uses CNN's to predict it and interprets the attention scores in a Granger-Causality sense to find causal drivers of the variable. The Granger-Causality method was evaluated in Ombadi et al. (2020), though a one-to-one comparison with results from TCDF is unfair, we comment on the similarities and differences we found with the former. Using Granger-Causality Ombadi et al. (2020) reported a high False Positive Rate, even with the small number of variables in their system. This highlights the problem with bivariate methods like PCC, Granger-Causality, etc., in a multivariate system. Where confounding (common cause) and autocorrelation severely affect the false positive discovery (Tuttle and Salvucci, 2017; Ombadi et al., 2020; Delforge et al., 2022). Comparatively, by adopting a multivariate approach and accounting for autocorrelation in its CNN architecture, TCDF was able to reduce the false positive discovery and performed the best (in FPR score) across all algorithms. However, TCDF failed to identify true positives, with the lowest Recall values (median Recall = 0.28). This may result from our inability to sufficiently train the CNN's, since the author's report F1 scores similar to the predecessor of PCMCI+, PCMCI (Nauta et al., 2019). We struggled to improve the Recall of TCDF by tuning its hyper-parameters. The process was particularly difficult owing to its high number of hyper-parameters, typical to any CNN architecture. This maybe since CNN's are better suited to predict spatial patterns and a different deep learning model like LSTM may yield better results (Kratzert et al., 2019).

### 4.1.3 VARLINGAM

VARLiNGAM produced one of the most contrasting results amongst the CD methods. It consistently had varying MCC scores in all climate classes. Though it scored Recall values similar to PCMCI+ (median Recall = 0.40), as reported previously (Assaad et al., 2022; Hasan et al., 2024; Runge, 2022). However, it was able to retrieve the non-linear link from plant canopy surface water into surface soil moisture which TCDF and PCMCI+ struggled to retrieve in any climate class. As suggested by

Hyvärinen et al. (2008), this may be a result of non-Gaussian errors obtained when VARLiNGAM uses linear models to fit the data.

#### 685 4.1.4 PCMCI+


PCMCI+ provided the most stable results across all climates in both Recall and MCC scores. Though, on average it could only identify 40 % of all the causal links (medain Recall = 0.41), which is a little lower than reported by Runge (2022). Further, the majority of the causal links in our adjacency matrix (Fig. 2b) are contemporaneous, thus the algorithm was able to retrieve these contemporaneous links, as suggested by Delforge et al. (2022). Ombadi et al. (2020) reported a similar Recall for the base algorithm, PC-alg, for similar lengths of data. This is expected since the dimensionality of our test was higher (4 in former, 27 in ours) and the lag relation up-to which the search was done (lag-1 in former, lag-1 and contemporaneous in ours). However, we obtained similar levels of false positives (FPR

alternative modules better suited in the context of their application. For example, as Runge et al. (2019b) and Runge (2022) suggested, if non-linear dependencies between variables are anticipated then the conditional independence tests of PCMCI+ can be done using various linear and non-linear methods. Similarly, if fewer lagged relations are expected in a system, then the initial estimate of the lagged adjacency matrix in VARLiNGAM can be calculated using a sparsity penalty (L1-norm of errors) instead of using a Ordinary Least Squares approach (L2-norm of errors).

Further, as mentioned by the authors of VARLiNGAM, the assumption of non-Gaussian errors makes it a unique tool in the family of Causal Discovery methods. We believe this makes it a valuable tool for CD in climate change scenarios, where non-stationarity in the distribution of hydro-meteorological variables is expected, which yields non-Gaussian model errors.

Although the objective of DYNOTEARS is to find a DAG, its theoretical implementation to find it as coefficients of a SVAR model allows for familiar interpretation. As SVAR models have long been used in Hydrology, particularly in forecasting models as ARMA, ARIMA and similar models. In the case of TCDF, by adopting the Granger-Causality framework, it avoids the fulfilment of the strong assumptions of Causal Sufficiency, Causal Markov assumption and Faithfulness on the data. This theoretically allows it to be applied in systems where the above assumptions cannot be satisfied.

#### 4.4 Limitations

There are several limitations to our work. First, as mentioned by Ombadi et al. (2020), causal interactions can evolve in time and different mechanisms can drive a process in different time period's. For example, they found wind speed as a causal driver of evapotranspiration during the summer season but no causal link was not identified in the winter season. This phenomenon of time-evolution of causal relations has been recognized in the literature and classified as an issue to be addressed when identifying causality in a time window or over the entire time-series (summary causal graph) (Ombadi et al., 2020; Assaad et al., 2022). To this end, we applied causal analysis over the entire time-series, thus focusing on causal interactions over the entire time-period. Second, we assumed stationarity of the time-series data, which is necessary for many statistical tools like Linear Regression, which forms the backbone of most of the methods discussed here. Third, by performing the analysis in a simulated environment we ensured Causal Sufficiency. As discussed above, this may not be possible in real-world applications. This leads to the fourth issue, where the number of variables in the system increases to very high numbers. This creates a problem for algorithms in terms of *i*) expanding the number of possible causal links that need to be evaluated (and eliminated if necessary), *ii*) convergence of methods and *iii*) computational time required. One could argue, for example, in the case of identifying the causal parents of surface soil moisture, that certain variables, like ground water storage, etc, could be excluded from the CD analysis as a direct cause-effect relationship is not expected between the variables. Thus, increasing the detection power of CD methods.

## 5 Conclusion

The science of cause-effect analysis has seen rapid development over recent years (Assaad et al., 2022; Gong et al., 2024). Inspired by Ombadi et al. (2020) and the premise of finding causally-related variables, we evaluated state-of-the-art methods

of Causal Discovery. While the former evaluated simpler methods of causal discovery in a simple lumped model, in the context of varying length of time-series, process and observational noise, here we evaluated *strictly* causal methods on output of a large, complex model. This allowed us to test the algorithms over a real-world like system while allowing the generating equations to be used as a benchmark of true causal relationships. We evaluated four theoretically distinct causal discovery methods (TCDF, VARLiNGAM, PCMCI+ and DYNOTEARS) that traverse the broad spectrum of causal discovery methods. We also overviewed some methods of representing causal relations via a graph and an adjacency matrix. By contrasting our results with PCC we exposed how bivariate and non-causal methods lead to inflated drivers.

We found the correlation based method PCC, identified the highest number of causal links, followed by DYNOTEARS. While other CD methods were able to *Recall* half or fewer causal links. However, CD methods were more effective at eliminating highly correlated but non-causal variables across climate types. By adopting multivariate frameworks with contemporaneous and lagged relations, CD methods were able to identify the correct order of lag relations amongst variables while eliminating multiple spurious correlations. This provided a parsimonious set of causal drivers of a process, potentially leading to a better process understanding and time-series prediction. To test the latter we identified drivers of surface soil moisture during normal conditions with PCC and CD methods, for time-series prediction. Evaluation during drought period showed CD-based machine learning models performed better, with higher performance scores and smaller variability, compared to correlation based models, highlighting the importance of finding causal drivers of a process. Finally, we discussed some caveats to applying CD methods in the real world and discussed how their assumptions and algorithms can be exploited to further retrieve causes of variables in hydrometeorological systems.

## 6 Code and Data Availability

All the data used in the analysis was downloaded from NASA Earth Data and can be publicly accessed at Li et al. (2018).

All the analysis and plots created were done using publicly available python libraries. The analysis was done using standard python libraries like numpy (Harris et al., 2020), scipy (Virtanen et al., 2020), jupyter notebooks (Project Jupyter et al., 2018) and plots were created using matplotlib (Hunter, 2007) and seaborn (Waskom, 2021). The code for the CD algorithms are publicly available. For TCDF, the code is available on GitHub TCDF https://github.com/M-Nauta/TCDF. For VARLiNGAM the lingam python package was used: https://github.com/cdt15/lingam, (Ikeuchi et al., 2023). For PCMCI+ the tigramite python package was used: https://github.com/jakobrunge/tigramite. For DYNOTEARS the causalnex python package was used: https://causalnex.readthedocs.io/en/latest/. The code to do the analysis and recreate the plots in this study are in https://github.com/lsmvivek/project\_ci\_eval.

# Appendix A: CLSM model: Description, modelling schemes and generating equations

#### A1 Model description

The CLSM model is composed of various routines to model different processes on the land surface. These routines are adopted or based on other works, Koster et al. (2000); Ducharne et al. (2000). The energy balance and canopy interception schemes are based on the MOSAIC LSM model, Koster and Suarez (1992); Koster et al. (1996 - 03??). The sub-surface moisture distribution is based on Clapp and Hornberger (1978). Using this distribution, the calculation of sub-surface storages and surface runoff generation is based on TOPMODEL from Beven and Kirkby (1979). Finally, the snow related simulations are based on Lynch-Stieglitz (1994). Since none of the grids selected in our analysis contained any snow variables (time-series was zeros), we do not discuss their governing equations.

The model is forced with nine meteorological forcings from a General Circulation Model. Precipitation, rainfall as fraction of precipitation, downward short-wave radiation flux, downward long-wave radiation, specific humidity, snow precipitation rate, air temperature, surface pressure and wind speed. Since these act as forcing to the model and are not affected by any feedback from it to the GCM, we consider these as independent variables and hence they do not have any causal drivers. Rather, these only act as causal drivers of other variables.

The model simulates various prognostic and internal variables, and outputs 33 hydro-meteorological variables. By ignoring the snow related variables we are left with 27 variables listed in Table A1. As mentioned above, eight of these are forcing variables (ignoring snow precipitation rate). Thus, we have a total of 20 dynamically simulated variables in our analysis that have causal drivers.

We found certain differences in the variables described in the original paper and the current version of model outputs. For example the papers describe how a TOPMODEL based bulk variable called Catchment-Deficit is used for sub-surface moisture distribution. However, the current output variables do not contain the same, rather a variable called Profile Soil Moisture is provided. Using visual inspection and literature review we consider Profile Soil Moisture to be the Catchment-Deficit term. Thus we make certain assumptions to overcome such issues wherever necessary to obtain all the governing equations and/or relations.

Further, apart from the forcing variables and dynamically generated variables, the model uses a host of static parameters which characterize the local distribution of vegetation and topography. These parameters affect the simulation of various variables, like vegetation indices affect the scaling of transpiration in zones between completely saturated and wilting zones. Thus in favour of simplicity, we do not mention the full complexity of the model and only mention model equations and/or describe the simulation routine, with the functional forms used to describe the causal relations amongst the variables, which form the reference truth in Fig. 2a.

## A2 Modelling schemes and generating equations

## **Canopy Interception**

The canopy interception reservoir has one incoming flux of precipitation and one outgoing flux of evaporation from it. Thus, it has the functional form:

$$CanopInt_t = f(Rainf_t, Rainf_t, ECanop_t)$$
(A1)

#### **Surface Runoff**

If the canopy interception reservoir is full after a precipitation event, the excess precipitation falls onto the land surface as through-fall precipitation. The model divides each spatial unit tile into three types based on the concurrent surface soil moisture. These are: completely saturated region, region at wilting point and the region between these two. The through-fall precipitation falling on the saturated region is immediately converted into storm surface runoff. While the through-fall precipitation on the latter two regions is scaled according to the surface soil moisture capacity. Thus, storm surface runoff is generated as:

$$Qs_t = \begin{cases} PT_t \cdot A_{\text{saturated}} & \text{if } Mse_t < 0 \\ PT_t \left( A_{\text{saturated}} + A_{\text{transpiration area}} \cdot \frac{Mse_t}{Mse \text{-} max_t} \right) & \text{if } Mse_t > 0 \end{cases}$$

where  $M_{se}$  is the surface excess given by surface soil moisture. This gives the functional form for Storm surface runoff as:

$$Qs_t = f(\mathsf{CanopInt}_t, \mathsf{SoilMoist-S}_t, \mathsf{Rainf}_t, \mathsf{Rainf-f}_t) \tag{A2}$$

#### **Surface and sub-surface storages**

In the CLSM model, surface and sub-surface soil moistures are simulated in two steps combining different modelling techniques. First a bulk catchment term called Catchment Deficit is calculated following Beven and Kirkby (1979). Second, this deficit is distributed across the layers of soil using the formulation in Clapp and Hornberger (1978). Then the local values of these storages are based on the antecedent conditions and the transfer of moisture between the vertical levels. The transfer of moisture from surface to atmosphere follows the equations in Koster and Suarez (1992) and sub-surface transfers are simulated using the TOPMODEL scheme from Beven and Kirkby (1979).

#### 825 Surface soil moisture

The surface soil moisture is modelled as a reservoir of moisture with incoming flux as precipitation (through-fall or direct) and outgoing flux as infiltration into root-zone soil moisture and evaporation from soil. At each time-step the model reduces the excess (or deficit) in this storage by percolating (or gained via capillary action) a fraction of the storage into the reservoir below. The amount percolated (or gained via capillary action) is proportional to the absolute storage, this adds a auto-correlation type relation in the variable. Thus its functional form is given as:

$$SoilMoist-S_t = f(Rainf_t, Rainf-f_t, SoilMoist-RZ_t, SoilMoist-P_t, CanopInt_t, ESoil_t, Evap_t, SoilMoist-S_{t-1})$$
(A3)

#### Root-zone soil moisture

The Root-zone soil moisture is modelled as a reservoir of moisture with incoming flux as moisture from Surface soil moisture and outgoing flux as percolation into groundwater and evaporation from soil. Root-zone soil moisture excess (or deficit) is also

reduced as proportional to the absolute value of storage. Thus its functional form is given as:

$$SoilMoist-RZ_t = f(SoilMoist-P_t, SoilMoist-P_t, GWS_t, ESoil_t, Evap_t, SoilMoist-RZ_{t-1})$$
(A4)

#### Profile soil moisture

The Profile soil moisture term was assumed to be related to the Catchment Moisture Deficit term in Koster et al. (2000). Catchment moisture deficit is the moisture required to completely saturate the sub-surface and bring the water table to near the surface. Thus it is updated according to the moisture storages in various sub-surface levels locally and summed over the entire catchment. Thus, its functional form is given as:

$$SoilMoist-P_t = f(SoilMoist-RZ_t, GWS_t, SoilMoist-P_{t-1})$$
(A5)

## Groundwater storage

Finally, the root zone moisture transfers the moisture to the water table in the groundwater. The groundwater acts as a reservoir for groundwater baseflow. Thus it has one incoming flux as percolation from Root-zone soil moisture and outgoing flux as baseflow discharge. Thus its functional form is given as:

$$GWS_t = f(SoilMoist-RZ_t, Qsb_t, GWS_{t-1})$$
(A6)

## **Baseflow-groundwater runoff**

The discharge from the groundwater storage is modelled as a non-linear function of the bulk catchment moisture deficit, the local water table depth and the mean water table depth of the catchment.

$$Qsb_t = \frac{K_s(surface)}{\nu} \cdot \exp(-\bar{x} - \nu \bar{d})$$

where  $\bar{d}$  and  $\bar{x}$  are the mean water table depth of the catchment and the local topography,  $K_s(surface)$  and  $\nu$  is the surface-saturated hydraulic conductivity and describes the exponential decay of the saturated hydraulic conductivity with depth. Thus, its functional form is given as:

$$Qsb_t = f(SoilMoist-P_t, GWS_t)$$
 (A7)

# Terrestrial water storage

Terrestrial water storage was not found to be mentioned in the original or supporting papers. Perhaps it was introduced much later in light of the GRACE mission. However, it has been defined in the product README document. Thus the TWS is defined following the same, with the functional form as:

$$\operatorname{Tws}_t = f(\operatorname{CanopInt}_t, \operatorname{GWS}_t, \operatorname{SoilMoist-P}_t,)$$
 (A8)

# **Evaporative Fluxes**

The total evaporation from the surface is modelled using a bulk effective resistance term,  $r_{eff}$ . It captures the effect of canopy resistance, aerodynamic resistance and bare soil resistance. Thus the total evaporation is given as:

$$Evap_t = \frac{\rho\epsilon}{\mathsf{Psurf-f}_t} \cdot \frac{es(T_c)_t - ea_t}{reff_t}$$

Therefore, its functional form is given as:

$$Evap_{t} = f(Tc_{t}, Tair_{t}, Qair_{t}, Psurf_{t}, ACond_{t}, WindSpeed_{t}, CanopInt_{t}, SoilMoist_{t}, SoilMoist_{t})$$
(A9)

Similar to the total evaporation the other evaporative fluxes are calculated as below:

$$\begin{split} \text{ECanop}_t &= min(\frac{\text{CanopInt}_t}{\Delta t}, \text{Evap}_t \cdot \frac{\text{CanopInt}_t}{\text{CanopInt-max}_t} \cdot \frac{r_a + r_{Tbs}}{r_a + \frac{\text{CanopInt}_t}{\text{CanopInt-max}_t} \cdot r_{Tbs}}) \\ & \text{Tveg}_t = (\text{Evap}_t - \text{ECanop}_t) \cdot (\frac{r_c}{r_c + r_{bs}}) \\ & \text{ESoil}_t = (\text{Evap}_t - \text{ECanop}_t) \cdot (\frac{r_{bs}}{r_c + r_{bs}}) \end{split}$$

where  $r_{bs}$  is the bare soil resistance to evaporation and  $r_c$  is the canopy resistance to transpiration, function of CanopInt<sub>t</sub> and WindSpeed-f<sub>t</sub>. Thus, we obtain the functional relationships as:

$$ECanop_{t} = f(Evap_{t}, CanopInt_{t}, SoilMoist-S_{t}, Wind-f_{t})$$
(A10)

$$ESoil_t = f(Evap_t, ECanop_t, CanopInt_t, SoilMoist-S_t, SoilMoist-RZ_t, Wind-f_t))$$
(A11)

$$\mathsf{Tveg}_t = f(\mathsf{Evap}_t, \mathsf{ECanop}_t, \mathsf{CanopInt}_t, \mathsf{SoilMoist-S}_t, \mathsf{Wind-f}_t)) \tag{A12}$$

# **Aerodynamic Conductance**

Aerodynamic conductance is defined on the data product website, as the measure of how effectively vapour flows through stomata openings, total leaf area and soil surface. It is the inverse of aerodynamic resistance. In the papers the effective aerodynamic resistance is defined as the summation of

$$ACond_t = f(Qair-f_t, Tair-f_t, Psurf-f_t, Wind-f_t)$$
 (A13)

# **Energy Balance equations and Vapour flux equation**

The above two equations are simultaneously solved using a first order linearization of terms to get the  $\delta T_c$  and  $e_a$  terms.

$$R_{sw-net} + R_{lw} = \frac{C_H \delta T_c}{\Delta t} + R_{lw} + H + \lambda E + G_D$$
$$E_{surface} = \frac{\rho E}{p_s} (e_s(T_c) - e_a)$$

These are then used to update the values ground heat flux, sensible heat flux and upward long-wave radiation fluxes as:

$$\begin{split} R^t_{lw} &= R^{t-1}_{lw} + \frac{\Delta R_{lw}}{\Delta t} | t \cdot \delta T^t_c \\ H^t &= H^t + \frac{\Delta H^{t-1}}{\Delta T^{t-1}_c} | t - 1 \cdot \delta T^t_c + \frac{\Delta H^t}{\Delta e^t_a} | t - 1 \cdot \delta e^t_a \\ G^t &= G^{t-1} + \frac{\Delta G^{t-1}}{\Delta T^{t-1}_c} | t - 1 \cdot \delta T^t_c \end{split}$$

This gives a functional form for the temperature of the surface/canopy system and the energy budget terms as:

390 
$$Tc_t = f(L_{lw-f}^t, S_{sw-f}^t, H^t, E^t, G_D^t, Qair-f_t, Tair-f_t, Psurf-f_t, T_c^{t-1})$$
 (A14)

$$H^{t} = f(H_{t-1}, T_{c}^{t}) \tag{A15}$$

$$G^{t} = f(G^{t-1}, T_c^t) (A16)$$

$$Qle_t = f(ECanop_t, Tveg_t, ESoil_t)$$
(A17)

$$R_{lw}^{t} = f(R_{lw-f}^{t}, T_{c}^{t}) \tag{A18}$$

$$L_{lw}^{t} = f(L_{lw-f}^{t}, T_{c}^{t}) \tag{A19}$$

## A3 Model forcings and simulation period

As described above, the model is forced with global meteorological forcing dataset from Princeton University Sheffield et al. (2006). The model simulations were initialized on simulation date January 1, 1948 using soil moisture and other state fields set at climatology. The total simulation period spans January 1, 1948 to 31 December, 2014. The models simulates the variables at 3-hourly intervals and provides the output at 3-hourly, daily and monthly temporal resolution. The data used in this study was from 1 January, 2002 till 31 December, 2014 at daily timescale resolution.

## **Appendix B: Algorithm settings**

Below we describe the algorithm parameters used and modifications applied for this study. Before running the analysis we standardized the data by subtracting its mean and dividing it by the standard deviation. Since the maximum lag in causal relations was set to be 1, we conducted the analysis by fixing the maximum lag parameters accordingly for testing.

## **PCC**






For identifying the drivers of a target variable, we found its Pearson's correlation coefficient with all the remaining variables in the system, both at contemporaneous time step and by creating their one-step-lagged time-series. Then, we selected only those variables as drivers where the p-value was smaller than 0.05 and absolute correlation coefficient greater than 0.2 (Wu and Chau, 2011). We chose a low correlation threshold to maximize the inclusion of potential causal drivers, while reducing the likelihood of retaining purely spurious associations via the significance threshold.

# **TCDF**

As summarized in Table 1, TCDF has six parameters. By reviewing the method paper, we chose the following parameters: maximum lag = 1, hidden layers = 0, kernel size = 4, dilation coefficient = 4, number of epochs = 1000, significance = 1 and learning rate = 0.005.

## PCMCI+

The parameters for PCMCI+ were minimum lag = 0 and maximum lag = 1. In addition, at the significance threshold  $\alpha_{pc} = 0.95$ , the conditional independence test was the partial correlation (ParCorr) from the tigramite package itself (see *Code availability*). Since the analysis data was from a perfect environment with no observational or process noise, this could

potentially lead to "perfect fitting" of various variables during the conditional independence tests. This leads to variables negating each other's effects, where in fact, both may be causal drivers (Ombadi et al., 2020). This is a case of violation of causal faithfulness, wherein the causal relations exist, however they cannot be recovered by conditional independence tests (Runge, 2022). Thus, to avoid such issues, we added randomly generated noise to the data before conducting the analysis. The noise was generated using a Gaussian random noise of  $N\left(0,(0.2\sigma)^2\right)$  where  $\sigma$  is the sample standard deviation of the time-series. We tested the algorithm in range of noise levels,  $0.1\sigma - 0.5\sigma$  and found the results to be qualitatively robust, giving confidence that the results are not an artifact from the level of noise. While this does not guarantee causal faithfulness, it allows us to create a non-deterministic system which is a condition for causal faithfulness. Further, it also makes the evaluation more comparable to realistic world scenarios where observational is present.

Further, since PCMCI+ models causal relations using a graph (the Directed Acyclic Graph), it allows prior knowledge to be injected into the graph. This allows the benefit of expert knowledge of the system (if any) to be utilised to find causal relations, instead of just relying on the algorithm. Thus, we explicitly removed incoming causal links into the eight forcing variables.

#### **VARLINGAM**






VARLiNGAM offers a single parameter to modify the algorithm, the maximum lag parameter which was selected as 1. A key assumption of VARLiNGAM is that error terms are non-Gaussian and mutually independent. However, majority of variables in our analysis showed Gaussian distribution. Thus, similar to PCMCI+, to satisfy this assumption and stabilize estimation, we added randomly generated noise sampled from a gamma distribution with scale factor = 0.1 and shape = 1. This introduces mild skewness while maintaining overall variance of the original time-series. We tested for varying levels of noise ranging between 0.1 to 0.3 scale factor and obtained qualitatively similar results. Thus, giving us confidence towards the robustness of the results towards the addition of noise.

Finally, the algorithm provides the modelled relations as the coefficients of a SVAR model (the adjacency matrix). Since these coefficients can take very small values, we applied a simple threshold of 0.1 for the contemporaneous adjacency matrix and 0.01 for the lagged adjacency matrix. The coefficients above these thresholds were considered to represent causal relations. We selected 0.1 for the contemporaneous adjacency matrix because we anticipated most links to be contemporaneous and hence have a larger coefficient in the SVAR model. Similarly, we expected fewer causal links in the lagged adjacency matrix, thus, selected a lower threshold of 0.01 for it.

# DYNOTEARS

For DYNOTEARS, we selected the sparsity penalty terms as  $\lambda_w = 0.001$  and  $\lambda_a = 0.01$ , while the maximum cyclicity allowed was  $h(\mathbf{W} = 0.01)$ , instead of 0 to allow for any converge errors issues. Then, similar to VARLINGAM, the causal relations in DYNOTEARS are represented by coefficients of the SVAR model. We select the  $\mathbf{W}_{threshold} = 0.01$  for the existence of a causal relation. Further, similar to PCMCI+, DYNOTEARS allows including the posterior probability of known causal relations to be included in the system, accordingly we restrict the causal relations incoming into the forcing variables.

## Appendix C: Machine learning model for time-series predictions

We used two machine learning models for prediction of variable time-series, support vector regression model and a feedforward neural network.

We implemented a Support Vector Regression (SVR) model with a radial basis function (RBF) kernel using the scikit-learn library (Pedregosa et al., 2011). Model hyperparameters were optimized using a grid search with three-fold cross-validation. The hyperparameter search space included the regularization parameter  $C \in \{0.1, 1, 10\}$ , the epsilon parameter  $\epsilon \in \{10^{-6}, 10^{-4}, 10^{-2}\}$ , and the kernel coefficient  $\gamma \in \{\text{scale}, \text{auto}\}$ . Model performance was evaluated using the coefficient of determination  $(R^2)$  as the scoring metric.

We implemented a feedforward neural network (FNN) using the Keras library (Chollet et al., 2015). The network consisted of three hidden layers with 32, 16, and 8 neurons, respectively. Rectified Linear Unit (ReLU) activation functions were used for the hidden layers, while the output layer employed a linear activation suitable for regression tasks. Light dropout regularization was applied to mitigate overfitting. The model was trained using the Adam optimizer with a learning rate of 0.001, minimizing the mean squared error (MSE) loss function.

The figures below show the performance and error metrics for the prediction of different variables, across increasing noise levels. The subplots show the scores in the training period and the difference in testing and training periods.

#### **Author Contributions**

VKY: Conceptualization, Data curation, Formal analysis, Methodology, Visualisation, Writing (original draft preparation), Writing (review and editing), MP: Supervision, Writing (review and editing), KF: Supervision, Writing (review and editing), DR: Supervision, Writing (review and editing), BDV: Conceptualization, Supervision, Writing (review and editing).

## **Competing Interests**

Some authors are members of the editorial board of journal Hydrology and Earth System Science.

### Acknowledgements

VKY gratefully acknowledges the financial support of Ministry of Education, Government of India through the PhD fellowship.

4 Additional support was provided by the Melbourne India Postgraduate Academy (MIPA) fellowship.

**Figure A1.** Performance and error metrics for surface soil moisture predictions over a grid in Ganga basin. Similar to 7 a)–c) and b)–d), the figures show the performance (and error metrics) during the training period and the difference in performance during the testing and training periods. Further, figures show the results across different levels of noise in the system. The machine learning model used for prediction is support vector regression model.

Figure A2. Similar to A1 but with a feedforward neural network model.

**Figure A3.** Similar to A1 but scores for prediction of surface storm runoff in the Ganga basin. Training period: 2000-01-01 to 2003-12-31, Testing period: 2004-01-01 to 2005-12-31.

**Figure A4.** Same as A3 but predicted using a feedforward neural network.

**Figure A5.** Similar to A1 but scores for prediction of transpiration in the Murray basin. Training period: 2007-01-01 to 2012-12-31, Testing period: 2005-01-01 to 2006-12-31.

**Figure A6.** Same as A5 but predicted using a feedforward neural network.

| Short Name        | Long Name                          | Unit       |
|-------------------|------------------------------------|------------|
| ACond_tavg        | Aerodynamic conductance            | m s-1      |
| AvgSurfT_tavg     | Average surface skin temperature   | K          |
| Qsb_tavg          | Baseflow-groundwater runoff        | kg m-2 s-1 |
| ECanop_tavg       | Canopy water evaporation           | kg m-2 s-1 |
| ESoil_tavg        | Direct evaporation from bare soil  | kg m-2 s-1 |
| Evap_tavg         | Evapotranspiration                 | kg m-2 s-1 |
| Qg_tavg           | Ground heat flux                   | W m-2      |
| GWS_tavg          | Ground water storage               | mm         |
| Qle_tavg          | Latent heat net flux               | W m-2      |
| Lwnet_tavg        | Net long-wave radiation flux       | W m-2      |
| Swnet_tavg        | Net short wave radiation flux      | W m-2      |
| CanopInt_tavg     | Plant canopy surface water         | kg m-2     |
| SoilMoist_P_tavg  | Profile soil moisture              | kg m-2     |
| Rainf_tavg*       | Rain precipitation rate            | kg m-2 s-1 |
| SoilMoist_RZ_tavg | Root zone soil moisture            | kg m-2     |
| Qh_tavg           | Sensible heat net flux             | W m-2      |
| SnowDepth_tavg    | Snow depth                         | m          |
| SWE_tavg          | Snow depth water equivalent        | kg m-2     |
| EvapSnow_tavg     | Snow evaporation                   | kg m-2 s-1 |
| Qsm_tavg          | Snow melt                          | kg m-2 s-1 |
| Snowf_tavg*       | Snow precipitation rate            | kg m-2 s-1 |
| SnowT_tavg        | Snow surface temperature           | K          |
| Qs_tavg           | Storm surface runoff               | kg m-2 s-1 |
| SoilMoist_S_tavg  | Surface soil moisture              | kg m-2     |
| TWS_tavg          | Terrestrial water storage          | mm         |
| TVeg_tavg         | Transpiration                      | kg m-2 s-1 |
| LWdown_f_tavg*    | Downward long-wave radiation flux  | W m-2      |
| SWdown_f_tavg*    | Downward short-wave radiation flux | W m-2      |
| Rainf_f_tavg*     | Total precipitation rate           | kg m-2 s-1 |
| Wind_f_tavg*      | Wind speed                         | m s-1      |
| Tair_f_tavg*      | Temperature                        | K          |
| Psurf_f_tavg*     | Surface pressure                   | Pa         |
| Qair_f_tavg*      | Specific humidity                  | kg kg-1    |

**Table A1.** Table describing the short and long names and units of simulated and forcing variables. \* marked variables are forcing variables. Obtained from the GLDAS model documentation Li et al. (2018).

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
