# Peer review of "Cause-effect discovery in Hydrometeorological Systems: Evaluation of Causal Discovery methods."

_EGUsphere, 2025_

## Referee Comment (RC1)

**Review of Manuscript**

**'Cause-effect discovery in Hydrometeorological Systems: Evaluation of Causal Discovery methods.'**

By V. Yadav et al.

Dear Editor,

I have reviewed the manuscript. My conclusions and comments are as follows:

**1. Scope**

The article is within the scope of HESS.

**2. Summary**

In their manuscript, the authors apply and evaluate several methods for causal discovery CD (TCDF, VARLiNGAM, PCMCI+, DYNOTEARS) and, as a benchmark, Pearson correlation coefficients (PCC) in a global hydro-meteorological virtual reality study (GLDAS-simulated timeseries of hydro-meteorological data, where through the process equation structure of the data-generating model the true cause-effect structures are known). They address the following research questions (here formulated in my own words): A) Can CD methods correctly identify process drivers?, B) Can CD methods separate process drivers from non-drivers?, C) Are there particular strengths and weaknesses of CD and correlation-based methods?, D) does parsimonious identification of process drivers provide advantages for model building in the sense of good out-of-sample prediction?.

To address their questions, the authors first define the true causal structure as an DAG-adjacency matrix constructed from the GLDAS process equations and define contingency-table-based measures of agreement between the true adjacency matrix and those returned by the CD and PCC methods: "Recall" for question A, "Matthews Correlation coefficient" for question B, "FPR-TPR" (false positive rate, true positive rate) plots for question C, temporally-out-of-sample performance of ML-based models using only the (random-noise infected) causal variables identified by the CD and PCC methods measured by $R^2$, NSE, $NSE_{mod}$, KGE, RMSE, MAE for question D. The CD and PCC methods are applied to a large set of globally distributed grid points covering several different Köppen-Geiger climate zones.

Analyzing the results of their experiments, the authors conclude with respect to their research questions A) that PCC identifies most of the true drivers, followed by DYNOTEARS, B) that PCC suffers from high FPR, and overall PCMCI+ and DYNOTEARS do best, C) CD methods outperform PCC mainly because of their ability to weed out false positives, D) ML-models using PCC-identified variables show higher training performance but sharper performance decrease in testing.

**3. Evaluation**

Overall, the manuscript is a timely and relevant contribution to the hydrological community, as it provides an introduction to causal discovery methods, which are not yet well known in the community but show large potential for hydrological system analysis and guiding model building. The manuscript is mainly well written, the experimental design is mainly adequate and the conclusions are mainly supported by the results. The structure does not yet optimally support the messages. In particular, there are several major and minor points that should be addressed

**Major points**

Points related to the definition of causality
- A cause-effect relation in the sense of this manuscript only exists between directly coupled nodes in a DAG. This differs from the colloquial interpretation, where indirect relations, e.g. between precipitation and streamflow, would also qualify as one. To help the reader, a clear definition of cause-effect relation (and how it differs from correlation) should be placed at the beginning of the manuscript, e.g. in the paragraph starting at Line 52.
- There is a fundamental ambiguity in the way how the "true" causal linkages are defined as those that can be extracted from model equations, as is done in the manuscript: Any multivariate equation of the form y = f (x1,x2,x3) can be re-expressed as a sequence of nested equations, e.g. y = f(x1, g(x2,x3)), or y = f(g(x1,x2),x3), etc. The choice of the nesting and sequential execution is more or less left to the preferences of the programmer. It will not change results, but it will change the resulting DAG, and with it what qualifies as a cause-effect relation, and what not, according to the definition in the manuscript. Any CD performed on a virtual reality derived from a set of process equations will suffer from this ambiguity, and I wonder to which degree this makes results useless.
- Most hydrological models use non-iterative numerical schemes, where a flux equation is followed by a state-updating equation. E.g. outflow from a linear reservoir is calculated as
  - $Q(t+1) = S(t)*k$
  - $S(t+1) = S(t) - Q(t+1) * dt$
    If I understood correctly from the manuscript, such a process equation structure cannot be represented by a DAG, because $Q=f(S)$ and $S = f(Q)$ and hence DAG-based CD methods cannot be applied. If correct, this would be a hindrance for the adoption of CD methods in hydrology, and should be mentioned as a limitation in the discussion or conclusion.
- In their study, the authors use an ideal situation where the full causal structure is perfectly known (see Fig. 2a) to evaluate CD methods. This is fine, but it will also be interesting to hydrologists what the potential of such methods is for system architecture identification. I.e. when only a few observables (usually forcing and target variables) are available, and the size and structure of the underlying system should be learned. I recommend adding a few words (and references) about this matter, e.g. in Sect. 4.4 or 5.
- Out of curiosity: In causality analysis, does the concept of an inhibitor exist? I.e. a variable that would effectively mask an existing causal relationship? For example, assume z = x + y. If x=1 and y=0, z=1. Also, for x=0 and y=1, z=1. So y effectively masks the causal dependency of z and x. This is not something to be addressed in the manuscript, but I would appreciate a reply.

Points related to the manuscript structure
- The results are often discussed separately for the different climate zones, or compared among them (e.g. Fig. 3, or Lines 510 pp). This is not reflected in the abstract and in the formulation of the research questions at the end of section 1. I recommend doing so.
- Line 109: Research Question (RQ) 4 is ambiguous: At this point in the manuscript, it is unclear what it means, and later in the manuscript it is used at two places: In Sect. 3.4 and Sect. 3.5. Sect 3.4 essentially addresses RQ 2 for a subsystem, so it should be labelled otherwise. Sect 3.5 addresses RQ 4. I suggest rephrasing RQ 4 to something like "Can CD methods help building parsimonious and robust hydrological models?"
- Sect. 2.5.1 - 2.5.4: Here the order of models differs from that in Table 1 and Sect 3.4. Please harmonize (I suggest keeping the order as in Table 1).
- Sect 2.5.7 should be a separate section 2.6, as it is topically separate from 2.5.1-2.5.6, which are all about CD methods.

- In Sect. 3, results are not only reported but also discussed. I suggest renaming it to "Results and Discussion". Also, I suggest mentioning at the beginning that the main substructure in this section is by the research questions RQ1-RQ4, and also reflecting this in the subsection headers. E.g. "3.1 RQ1: Can CD methods …"
- Sect. 3.4.6 is a summary statement, and would be better placed later in the manuscript
- Sect. 4: Here the main structure differs from that in Sect. 3. I recommend merging the two, structuring them along the RQs, and moving any parts that go beyond the immediate results and discussion of the experiment to the last section, which could then be named "Summary, Conclusions and Outlook"

Points related to manuscript content
- I really like the experiments and analyses related to RQs A-C, but the experiment for RQ D is not convincing. Why should a random error imposed on the identified drivers help distinguishing robust models with good generalization from non-robust models with poor generalization? From the information inequality we know that "information does not hurt", i.e. adding predictors will never worsen predictions. This is always true, and it shows in the superior performance of the PCC-based model in training, but the catch is that with increasing number of predictors, the curse of dimensionality kicks in, the available sample quickly becomes non-representative, overfitting occurs, and out-of-sample performance will drop. So a convincing demonstration of "CD returns fewer drivers than PCC, therefore the training data are more representative, therefore out-of-sample prediction is better" must include sample size. I recommend doing the following: Learn the different ML models (with input as selected by PCC and the CD models) on differently sized training data (from very small to the entire period 2000-2003) and apply on 2004. The CD-based models should do much better for small training sample sizes than the PCC-based.
- Sect 2.5 A description of how PCC was used in the study is missing. Please add, comparable to the descriptions of the CD methods in Sects 2.5.1-2.5.4
- In Sect. 3.5, results are reported for the ANN approach, but in the Appendix C SVR results are also shown. Consider removing them if not relevant, or also discuss them.

**Minor points**
- Line 81: I assume you mean "Time series produced by hydrological systems are …"
- Line 176: causes in rows and effect in columns: This is opposite to what's shown in Fig. 1.
- Line 211: Slightly misleading. I suggest rephrasing to: "We surveyed various models and their outputs with the …"
- Line 211 pp: Was global coverage also a criterion? If yes please mention.
- Fig. 2:
  - In a) ,please make clear what are the causes and what are the effects
  - b) - f): Pictures and legend to not match: It is 5 regions, in the text it says six major river basins. Also, the stars in the maps, which I assume depict the grid points, are not nine per map (as stated in the legend).
  - Later in the manuscript, in Fig. 3, are shown for 9 Köppen-Geiger classes and 9 river basins, which does not match the 5 plots in Fig. 2. Please harmonize.
- Line 243: what's k in the equation?
- Line 468: Why did you select the Ganga basin? Please justify
- Fig. 5: Unclear which subplot is for which method. Please add labels
- Fig. 7
  - The metrics are not explained. E.g. what is $NSE_{mod}$?
  - d): If I interpret correctly, testing performance for the CD models expressed by RMSE and MAE is better than for the training period. Is this correct? It could be because testing is in

a dry year, where soil moisture is generally lower, therefore absolute errors are also lower. In any case, please add an explanation to the text.

- o The requested changes might be obsolete if the figure is completely changed (see my comment on RQ D)
- Line 729: remove "not"
- 788: Why are the snow-related variables ignored? Please explain.

Yours sincerely,
Uwe Ehret

---

## Referee Comment (RC2)

The authors attempt to use "real-world data" (GLDAS) to systematically evaluate the effectiveness of different causal inference methods. I believe this is an important step for the development of the causal discovery field, which has long lacked benchmark testbeds based on realistic and complex systems for a rigorous evaluation of algorithmic performance. This absence of realistic benchmarks has made me skeptical about the practical usefulness of different causal discovery algorithms—especially those built on different assumptions—when applied under real-world, highly complex conditions. From this perspective, I think this work represents a valuable and meaningful attempt in this direction. However, there are several MAJOR issues in the study that, in my view, require further clarification and discussion.

**Concerns with "true" causality matrix**

When constructing the "true" causal adjacency matrix, the authors appear to consider only a one-day lagged causal effect. While this assumption may be reasonable for some fast-response flux variables (e.g., surface energy fluxes), many land-surface variables are known to exhibit pronounced long-term memory, such as soil moisture, groundwater storage, and snowpack. These memory effects typically influence the current state through multi-step Markov processes, rather than solely through the immediately preceding time step. It is worth noting that nearly all causal discovery methods evaluated in this study are, in principle, capable of explicitly accounting for multi-lag causal relationships. Therefore, assuming a uniform one-day lag in the reference "true" adjacency matrix may limit the realism of the benchmark and potentially affect the fairness of the evaluation.

**Concerns with the selection of causality discovery methods**

The authors compare four causal discovery algorithms that originate from distinct methodological paradigms. However, the manuscript does not sufficiently justify why these four specific algorithms were selected over other causal inference methods that are more commonly used in the hydrometeorological community. While PCMCI and its variants have seen increasing adoption in atmospheric and hydrological studies, widely used approaches such as CCM and Granger causality have not been directly included in the comparison. I therefore recommend that the authors provide a more explicit and systematic rationale for their choice of algorithms.

**Concerns with the comparison of different methods**

The manuscript presents an extensive set of quantitative analyses, using multiple statistical metrics to demonstrate the overall performance of different causal discovery methods. However, the connection between the discovered causal structures and real hydrometeorological processes is not sufficiently explored. This will bring some misleading result. For example, given that causal graphs are subject to Markov

equivalence, different DAG structures may yield similar performance.

In this regard, the manuscript would benefit from the inclusion of more concrete case studies. For example, what specific driving variables are identified by CD methods in different basins or climate regimes, and how do these compare with those selected by PCC? Are the identified drivers consistent with known physical mechanisms governing land–atmosphere interactions, hydrological processes, or energy balance? Conversely, which suspicious or physically implausible links are removed by CD methods relative to correlation-based approaches? Moreover, the authors may consider presenting spatial patterns of the inferred causal drivers, for example by mapping the dominant drivers across grid points within a given basin or region. Such spatially explicit analyses would offer a more intuitive and diagnostic perspective on method performance.

**Concerns with the result of predicting time-series**

The authors attempt to evaluate different causal discovery methods by using the variables selected by each method to drive machine learning models, and then comparing predictive performance as a proxy for causal effectiveness. However, I believe the resulting conclusions require more careful interpretation.

First, the number of features selected by PCC and by the CD methods differs substantially. Under identical training data, training protocols, and hyperparameter settings, models with a larger number of input features generally have a higher risk of overfitting and poorer generalization performance. As such, the reported differences in predictive skill may primarily reflect differences in feature dimensionality rather than the intrinsic quality or causal relevance of the selected predictors.

Second, soil moisture is a state variable with strong temporal memory. Its current value is typically highly and approximately linearly correlated with its lag-1 state, which, in practice, already contains most of the predictive information about the system. From my experience, introducing complex models or large sets of external predictors can sometimes degrade this physically consistent memory structure, leading to unstable or nonphysical mappings.

If the authors wish to retain a prediction-based comparison, I strongly recommend adopting a more controlled and interpretable experimental design. For example, this could include: (i) enforcing the same number of input features across different methods (e.g., using only the top-k ranked predictors), (ii) ensuring comparable model capacity or parameter counts across experiments, and (iii) explicitly accounting for the role of lag-1 soil moisture as a baseline or control predictor.

**Specific comments**

Line 201: The use of the Matthews Correlation Coefficient (MCC) requires further explanation, as it is not a commonly used metric in this context. In particular, the authors

should clarify its interpretation and why it is more appropriate than standard metrics under the highly imbalanced adjacency matrix setting.

Line 250: Incorporating acyclicity as a soft constraint in the loss function does not strictly guarantee a DAG solution, and such soft constraints can fail in practice, especially under finite-sample conditions or suboptimal hyperparameter choices.

Line 266: Eq.(9) appear before Eq.(8).

Line 569-470: The manuscript does not clearly justify why these specific years were selected for model training and validation.

Line 475: I do not consider the addition of random noise alone to constitute a more realistic scenario, nor does it meaningfully test robustness, since many error assumptions are already based on Gaussian noise.